# GALNTL5 binds GalNAc and is required for migration through the uterotubal junction and sperm-zona pellucida binding

Taichi Noda [1,2,3,11] ✉, Reika Uriu [1,11], Daisuke Mashiko [3], Hina Shinohara[1], Yongcun Qu[4], Ayumu Taira[1], Ryan M. Matzuk [5], Duri Tahala [1], Motochika Nakano[1], Kimi Araki [1,6], Zhifeng Yu [5], Ying Zhang [7], Martin M. Matzuk [5] ✉ & Masahito Ikawa [3,8,9,10] ✉

More than 20 genes expressed in the male reproductive tract have been identified as essential factors for sperm migration to and through the utero-tubal junction (UTJ), and they are divided into ADAM3-dependent and ADAM3-independent pathways. In parallel, sperm having UTJ migration defects also show impaired binding to the zona pellucida (ZP). Herein, we demonstrate that knockout of *Galntl5*, encoding a sperm surface protein, causes impaired sperm binding with the UTJ and ZP, and null males have severe infertility. GALNTL5 appreciably disappears in sperm lacking *Adam3* or *Lypd4*, required for ADAM3-dependent and ADAM3-independent pathways, and GALNTL5 binds to *N*-acetylgalactosamine (GalNAc) distributed on the UTJ and ZP. Blockage of GalNAc decreases the number of sperm binding to the UTJ and ZP. Thus, we unveil that GALNTL5 is a responsible factor for UTJ migration and sperm-ZP binding, and that sperm bind to the UTJ and ZP through interaction of GALNTL5 and GalNAc.

Sperm exit the testis, acquire fertilizing ability during the epididymal transition, and subsequently fertilize oocytes in the oviductal ampulla of the female reproductive tract. For fertilization to occur, ejaculated sperm must have the capability to migrate from the uterus into the oviduct, release sperm acrosome enzymes, and alter their motility pattern. During the last stages of the fertilization process, sperm bind to the zona pellucida (ZP), the glycoprotein layer surrounding the eggs. The factors responsible for sperm-ZP binding have been a constant search since Lillie first visualized the interaction between sperm and oocyte components more than 100 years ago[1].

Mouse ZP is composed of three heavily glycosylated proteins called ZP1, ZP2, and ZP3[2,3]. ZP3 appears to be more important than ZP1 and ZP2 for sperm-ZP binding because ZP3 protein purified from unfertilized eggs inhibits sperm-ZP binding[4]. Because oocytes from *Zp3* knockout (KO) mice lack a ZP[5], the physiological functions of ZP3 by itself cannot be analyzed. Further, after digestion of ZP3 with Pronase (a mixture of proteases), the small ZP3 glycopeptides continue to

[1]Institute of Resource Development and Analysis, Kumamoto University, 2-2-1 Honjo, Chuo-ku, Kumamoto, Japan. [2]Priority Organization for Innovation and Excellence, Kumamoto University, 2-39-1 Kurokami, Chuo-ku, Kumamoto, Japan. [3]Research Institute for Microbial Diseases, The University of Osaka, 3-1 Yamadaoka, Suita, Osaka, Japan. [4]Institute of Artificial Intelligence in Sports (IAIS), Capital University of Physical Education and Sports, Beijing, P. R. China. [5]Center for Drug Discovery and Department of Pathology & Immunology, Baylor College of Medicine, One Baylor Plaza, Houston, TX, USA. [6]Center for Metabolic Regulation of Healthy Aging, Kumamoto University, 1-1-1, Honjo, Chuo-ku, Kumamoto, Japan. [7]The Key Laboratory of Cell Proliferation and Regulation Biology, Ministry of Education, College of Life Sciences, Beijing Normal University, Beijing, China. [8]The Institute of Medical Science, The University of Tokyo, 4-6-1 Shirokanedai, Minato-ku, Tokyo, Japan. [9]Center for Infectious Disease Education and Research (CiDER), The University of Osaka, Suita, Osaka, Japan. [10]Center for Advanced Modalities and DDS (CAMaD), The University of Osaka, Suita, Osaka, Japan. [11]These authors contributed equally: Taichi Noda, Reika Uriu. ✉e-mail: noda-t@kumamoto-u.ac.jp; mmatzuk@bcm.edu; ikawa@biken.osaka-u.ac.jp

inhibit sperm-ZP binding[6], indicating that there is a sperm protein that binds the ZP3 protein-carbohydrate unit. ZP3 contains both N-linked and O-linked oligosaccharides[7–9]. The removal of all N-glycans using Endo-β-N-acetyl-D-glucosaminidase F (Endo-F) and the disruption of all branched antennae that contain galactose, fucose, sialic acid, and N-acetyl glucosamine residues on N-glycans using the mannoside acetylglucosaminyltransferase 1 (*Mgat1*) KO mice did not affect sperm-ZP binding or fertilization[10,11], while the removal of O-glycans from ZP using alkaline β-elimination decreased the number of sperm bound to the ZP[10]. These results suggest that O-linked oligosaccharides on the ZP are more critical for sperm-ZP binding than N-linked oligosaccharides.

O-linked oligosaccharides in mammals are classified predominantly into four core glycan structures (cores 1–4). So far, core-3-derived and core-4-derived O-glycans have not been detected by mass spectrometry analysis of mouse ZP3[12–14]. The physiological functions of core-1-derived and core-2-derived O-glycans for sperm binding have been analyzed using genetically modified mice, but the deletion of *T-syn* (core 1 β1,3-galactosyltransferase 1) and N-acetylglucosaminyltransferase (C2GnT-L and Gcnt1), which are required for the syntheses of core-1 and core-2-derived O-glycans, failed to affect female fertility and sperm binding number[15,16]. Thus, the roles of O-linked oligosaccharides on the ZP in sperm binding remain unknown.

Some sperm proteins, such as acrosin and PH-20 (also known as SPAM1), have been suggested as ZP3 receptor candidates using in vitro experiments[17]. Of these, β1,4-galactosyltransferase (known as GalT and β4GalT-1) was initially the most promising ZP3 receptor because GalT-bound ZP3 O-linked oligosaccharides caused the subsequent acrosome reaction through the activated signaling cascades of pertussis toxin-sensitive heterotrimeric guanine nucleotide-binding protein[18,19]; however, later studies revealed that sperm lacking GalT could bind to ZP and fertilize eggs[20,21]. SP56 (also called ZP3 receptor) was another ZP3 receptor candidate, as its alternative name implies[22–24]; however, *Sp56* KO sperm could bind to ZP and KO male mice were fertile[25], ruling it out as the sole ZP3 receptor. Thus, the candidates suggested by in vitro experiments were mostly dispensable for sperm-ZP binding. However, our team previously identified Calmegin (*Clgn*) as a testis-specific chaperone, essential for sperm-ZP binding[26]. Further studies revealed that CLGN is required for the heterodimerization of a disintegrin and metallopeptidase domain (ADAM) 1a and ADAM2, as well as the subsequent maturation of ADAM3[27,28]. Sperm lacking ZP binding ability also exhibit impaired migration through the utero-tubal junction (UTJ)[29], indicating that shared mechanisms exist for sperm-ZP binding and UTJ migration. So far, about 20 genes abundantly expressed in the testis and epididymis, such as a testicular isoform of angiotensin-converting enzyme (*t-Ace*) and *Adam3*, have been identified as essential factors for both sperm-ZP binding and UTJ migration[27,30–33]. Sperm with a knockout of most of these genes have ADAM3 loss, but there are exceptions. For example, our teams reported that ADAM3 persists in sperm lacking post GPI attachment to proteins 1 (*Pgap1*), lymphocyte antigen 6 complex, locus K (*Ly6k*), or Ly6/Plaur domain containing 4 (*Lypd4*), despite the findings that these KO sperm show a UTJ migration defect and impaired ZP binding[34–36]. Specifically, PGAP1 has a function as a GPI inositol-deacylase that removes the palmitate from inositol in the endoplasmic reticulum, suggesting that it is unlikely that PGAP1 is directly involved in the sperm migration into the UTJ and sperm-ZP binding[37,38]. LY6K, a GPI-anchored protein, interacts with testis-expressed gene 101 (TEX101), which is a substrate of t-ACE in testicular germ cells[35,39]. The role of LYPD4 in testicular germ cells and sperm in UTJ migration and ZP binding remains unclear, but this protein is present in *Adam3* KO sperm[36]. Thus, we conclude that other sperm factors, rather than ADAM3, which is not conserved in humans, are responsible for sperm-ZP binding and sperm migration through the UTJ.

UDP-N-acetyl-alpha-D-galactosamine: polypeptide N-acetylgalactosaminyltransferase-like 5 (GALNTL5) (also known as pp-GalNAc-T19[40], NCBI Reference Sequence: NP_660335.2) was identified as one of the N-acetylgalactosamine (GalNAc) transferase-like proteins required for the synthesis of GalNAc-linked glycans (also known as mucin-type O-glycans)[40]. Later studies revealed that GALNTL5 is abundantly expressed in human testis, and that GALNTL5 does not have GalNAc transferase enzyme activity in vitro[41–43]. Thus, GALNTL5 functions other than GalNAc transferase activity were analyzed using genetically modified mice; however, the *Galntl5* KO phenotype published by Takasaki et al.[42] is different from the International Mouse Phenotyping Consortium (IMPC) database. The IMPC database suggests that the male fertility of *Galntl5* KO mice is comparable to the control, while Takasaki et al.[42] published that a heterozygous null mutation of *Galntl5* resulted in infertility due to impaired sperm motility and morphology. To resolve this enigma and establish the physiological functions of GALNTL5 in male fertility, we created four independent lines lacking mouse *Galntl5* using CRISPR/Cas9. We reveal that not heterozygous but homozygous null mutations of *Galntl5* are almost exclusively male sterile and that GALNTL5 regulates sperm migration into the UTJ and sperm-ZP binding through the binding with GalNAc in the UTJ and on the ZP. We conclude that sperm GALNTL5 is a long thought responsible factor for sperm-ZP binding and sperm migration into the UTJ.

## Results

### The *Galntl5* locus encodes two transcripts

We first examined *Galntl5* mRNA expression by PCR using cDNAs from mouse multi-tissues. Two alternative splice variants are transcribed from the genomic sequence coding *Galntl5* (Fig. 1a), so we used a primer set to amplify both variants for the multi-tissue expression analysis. *Galntl5* mRNA was specifically expressed in the mouse testis (Fig. 1b), and its expression was detected 20 days after birth or later (Fig. 1c), indicating that *Galntl5* mRNA begins to be expressed in secondary spermatocytes and later stages. Also, human GALNTL5 is predominantly detected in the testis (Fig. 1d), consistent with the result of the real-time PCR in the previous paper[42].

### Loss of GALNTL5 causes fertility defects in male mice

To examine the physiological functions of mouse GALNTL5, we generated *Galntl5* mutant mice by injecting a guide RNA (gRNA)/Cas9 expressing plasmid into eggs to delete both alternative splice variants (Fig. 1a). By mating F0 mutants with wild-type (WT) mice, we established two mutant lines; enzyme mutation (*em*) *1* and *em2* disrupt 17 and 25 nucleotides in exon 2 of the *Galntl5* variants, respectively, leading to frameshift mutations (Fig. 1e and Supplementary Fig. 1a–c). By intercrossing of heterozygous (Het) mutants, we obtained *Galntl5* homozygous mutant (KO) mice. To check the disruption of GALNTL5 proteins in KO mice, we performed western blot analysis using extracts of testicular germ cells (TGC) and sperm and antibodies to recognize both termini of GALNTL5 (Supplementary Fig. 1d). We detected the doublet bands corresponding with predicted molecular sizes of variants 1 and 2 (~50 kDa and ~46 kDa) in the TGC extract in the control group, while a single band at ~37 kDa in sperm extracts was detected in the control group only when we used the antibody to recognize the C-terminus (Fig. 1f and Supplementary Fig. 2), indicating that sperm GALNTL5 lacks the N-terminal region. In the extracts of the TGC and sperm from KO mice, these bands disappeared, indicating that the longer TGC variants and the shorter sperm variant of GALNTL5 are disrupted in KO mice (Fig. 1f and Supplementary Fig. 2). To examine the timing when GALNTL5 is processed from ~50 kDa to ~37 kDa, we collected testicular sperm (TS) and sperm from caput, corpus, and cauda epididymides. As shown in Supplementary Fig. 3, when we used the N-terminus antibody, we could not detect the longer forms in

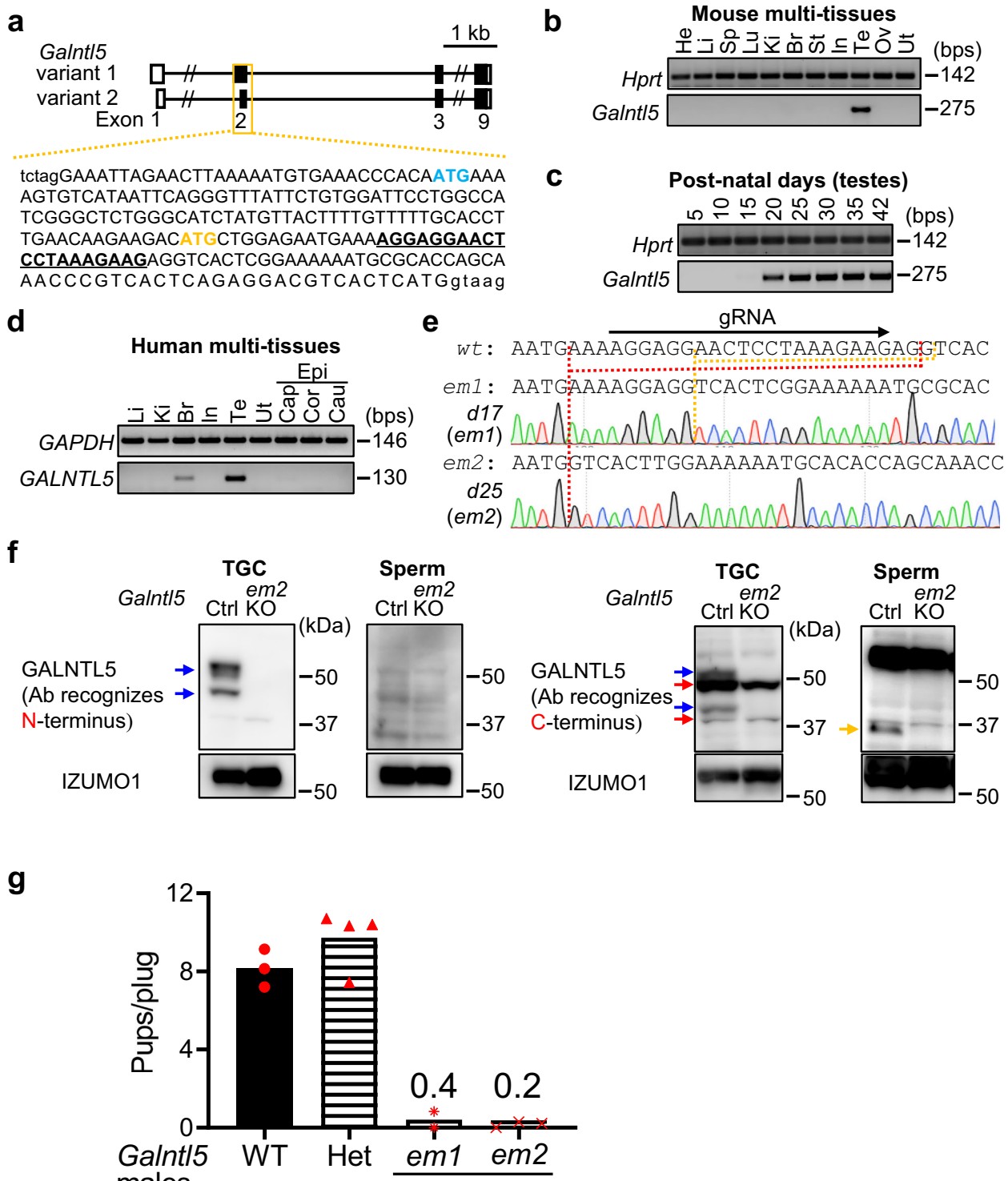

epididymal sperm but only in TGC and TS. Using the C-terminus antibody, we barely found a specific signal in the TS due to the poor reactivity of the antibody, but we could detect the longer form in TGC and the shorter form in sperm after the caput epididymis. Based on these data, we speculated that the longer form of GALNTL5 is cleaved after entering the caput epididymis by some protease(s).

To reveal the effects of disruption of GALNTL5 on male fertility, Het and KO male mice were mated with WT females. The fertility of Het males was comparable to WT males. However, females mated with KO males rarely delivered pups, indicating that *Galntl5* KO males are nearly sterile [pups/plug: 8.2 ± 1.0 (WT), 9.7 ± 1.5 (*em1* Het or *em2* Het), 0.4 (*em1* KO), 0.2 ± 0.2 (*em2* KO)] (Fig. 1g and Supplementary Table 1). Thus, despite creating two independent null alleles, we did not observe

**Fig. 1 | Male mice disrupted for both variants of *Galntl5* are nearly sterile.**
**a** gRNA design. There are two variants for mouse *Galntl5*. Blue-colored and yellow-colored "ATG" show the 1st Methionine for variants 1 and 2, respectively. The underlined sequence was used as the gRNA. Upper-case and lower-case letters show exon and intron regions, respectively. **b** Detection of *Galntl5* mRNA using mouse multi-tissues. Hypoxanthine guanine phosphoribosyl transferase (*Hprt*) is used as the control. The data reproducibility was checked by two technical replicates. Br: brain, He: heart, In: intestine, Ki: kidney, Li: liver, Lu: lung, Ov: ovary, Sp: spleen, St: stomach, Te: testis, and Ut: uterus. **c** Detection of *Galntl5* mRNA using mouse testes at 5–42 days of birth. The data reproducibility was checked by two technical replicates. **d** Detection of *GALNTL5* mRNA using human multi-tissues. Glyceraldehyde-3-phosphate dehydrogenase (*GAPDH*) is used as the control. The data reproducibility was checked by two technical replicates. Cap: caput, Cau: cauda, Cor: corpus, Epi: epididymis. **e** Direct sequencing. The 17 and 25 nucleotides

(nts) were deleted in the *em1* and *em2* alleles of *Galntl5*, leading to the appearance of a premature stop codon by the frameshift mutation (see Supplementary Fig. 1c). *wt*: wild-type, *em*: enzyme mutation. **f** Detection of GALNTL5 proteins in testicular germ cells (TGC) and sperm. We used antibodies to recognize N- or C-termini of mouse GALNTL5 (Supplementary Fig. 1d and Supplementary Table 4). In the control (ctrl) TGC, two bands were detected at the molecular sizes of variants 1 and 2 (~50 kDa and ~46 kDa, respectively) (see blue arrows). Using the C-terminus antibody, we also found non-specific bands of ~50 and ~37 kDa (red arrows) in both control and KO TGCs. In the ctrl sperm, GALNTL5 was detected at ~37 kDa only when we used an antibody to recognize the C-terminus (a yellow arrow). These specific bands disappeared in *Galntl5* KO TGC and sperm. IZUMO1 was used as the loading control. The data reproducibility was checked by four biological replicates. **g** Male fertility. The females mated with *Galntl5* KO males lacking variants 1 and 2 (v1 + v2) rarely delivered pups.

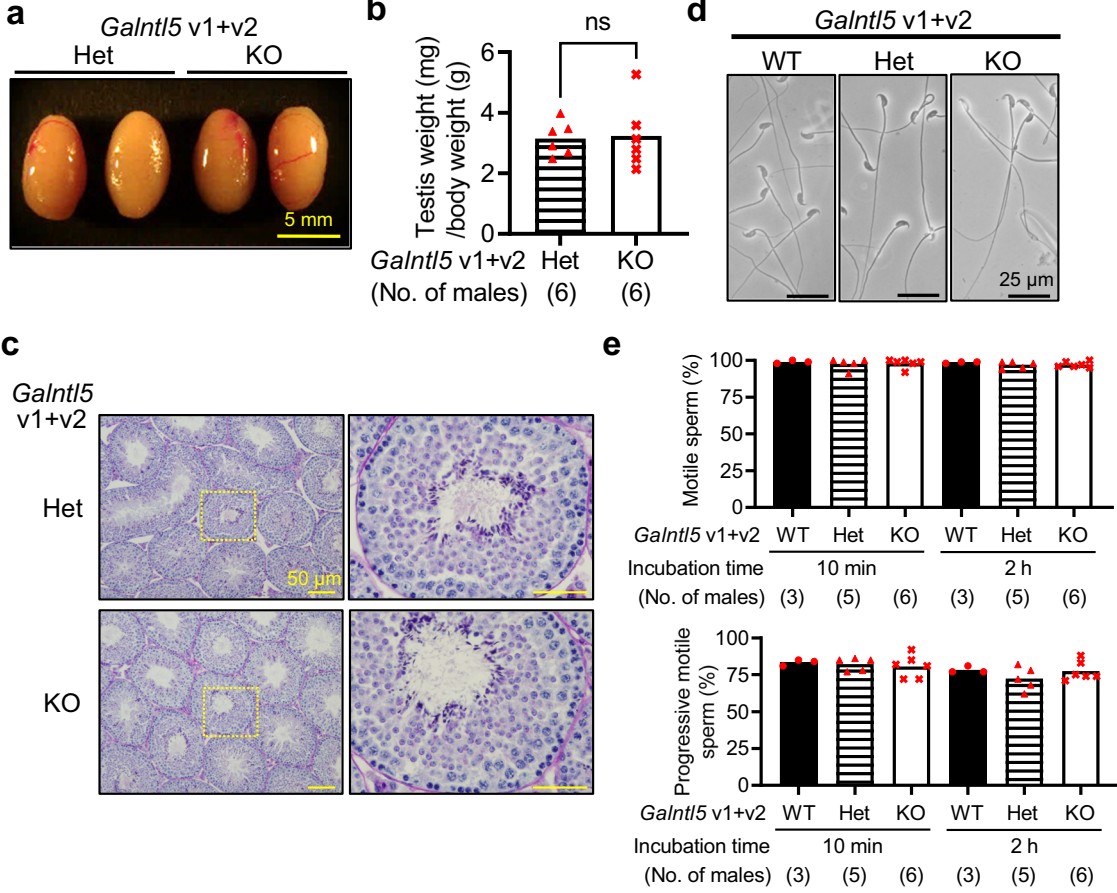

**Fig. 2 | Spermatogenesis and sperm morphology, and motility in Het and KO males of *Galntl5* variants 1 and 2 are normal. a** Testis morphology. **b** Testis weight. There was no difference in the testis weight between *Galntl5* Het [3.15 ± 0.56 (*em2/wt*)] and KO mice [3.2 ± 1.1 (*em2/em2*)] (Mann–Whitney test, *p* = 0.94). ns: not significant. **c** Histological analysis of testes with Periodic Acid Schiff (PAS)-Hematoxylin staining. The area dotted by the yellow color was magnified (right panels). The data reproducibility was checked by two biological replicates. **d** Sperm

morphology. The microscopic observation of sperm from *Galntl5* Het and KO mice was comparable to WT sperm. The data reproducibility was checked by two biological replicates. **e** Sperm motility analysis with CASA. After 10- and 120-min incubation, each parameter of the sperm motility was measured. There were no differences among the groups [Kruskal–Wallis test, sperm motility: *p* = 0.99 (10 min), and *p* = 0.55 (2 h), sperm progressive motility: *p* = 0.83 (10 min), and *p* = 0.50 (2 h)].

haploinsufficiency in our heterozygous mutant mice, unlike a previous paper[42].

### *Galntl5* KO sperm have a defect in passage through the UTJ
As there is no difference in male fertility between both KO mutant lines (*Galntl5*[em1] and *Galntl5*[em2] KO) (Fig. 1g), we used *Galntl5*[em2] mutants for further experiments. To understand why *Galntl5* KO males show severe subfertility and sterility, we examined spermatogenesis and sperm

motility. The gross morphology and weight of *Galntl5* KO testes were comparable to the control [testis weight/body weight: 3.2 ± 0.6 (Het), 3.2 ± 1.1 (KO)] (Fig. 2a, b). We observed spermatogenesis by microscopic observation using Hematoxylin and Periodic acid-Schiff staining, but we could not find an obvious abnormality (Fig. 2c). Furthermore, sperm morphology from *Galntl5* Het and KO mice was comparable to WT sperm (Fig. 2d), and we could not detect any defects in sperm motility parameters (Fig. 2e), unlike a previous publication[42].

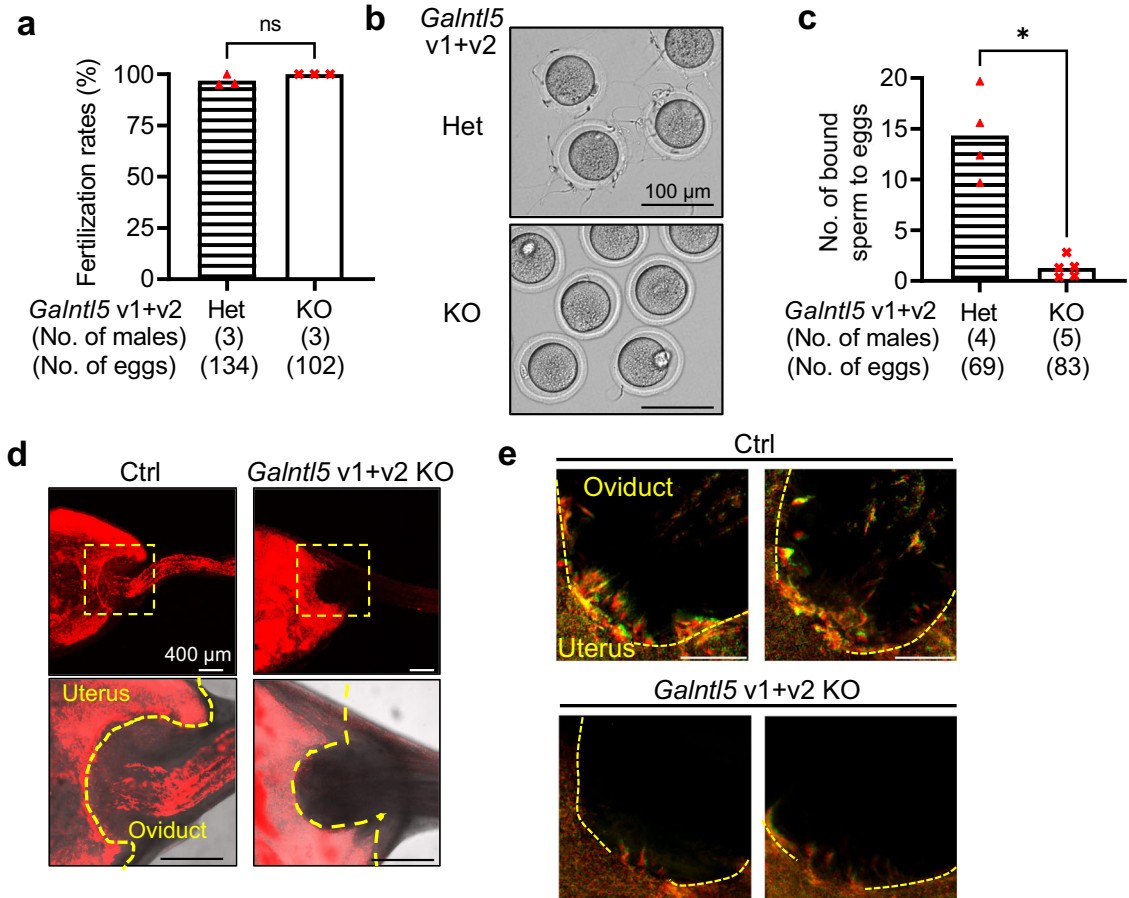

**Fig. 3 | *Galntl5* KO sperm lacking variants 1 and 2 show defects in ZP binding and migration into the oviduct. a** Fertilization rates with cumulus-intact eggs. Sperm from *Galntl5* KO mice could fertilize eggs at a comparable level as sperm from *Galntl5* Het mice [96.9 ± 2.7% (*em2/wt*), 100.0% (*em2/em2*)] (Mann–Whitney test, *p* = 0.40). ns: not significant. **b** Observation of sperm binding to the ZP. **c** Number of sperm bound to the ZP. Sperm from *Galntl5* KO mice hardly bind to the ZP [sperm number/egg: 14.3 ± 4.3 (*em2/wt*), 1.3 ± 1.0 (*em2/em2*)] (Mann–Whitney test, *p* = 0.02). **p* < 0.05. **d** Observation of sperm behavior in the

female reproductive tract. Though the fluorescence-labeled sperm from *Galntl5* KO mice abundantly exist in the uterus, these KO sperm hardly pass through the UTJ. The area dotted by the yellow color was magnified (lower panels). The data reproducibility was checked by five biological replicates. **e** Observation of sperm binding at the UTJ. The fluorescence-labeled WT sperm bind to the UTJ, but *Galntl5* KO sperm hardly bind to the UTJ. Scale bars were 200 μm. Yellow-dotted lines show the border between the uterus and the oviduct. The data reproducibility was checked by two biological replicates (WT) and three biological replicates (KO).

To assess the sperm fertilizing ability, we incubated *Galntl5* KO sperm with cumulus-intact and cumulus-free eggs. *Galntl5* KO sperm efficiently fertilized cumulus-intact eggs [fertilization rates: 96.9 ± 2.7% (Het), 100% (KO)] (Fig. 3a), indicating that *Galntl5* KO sperm show normal fertilizing ability in vitro. Furthermore, when these fertilized eggs were transferred into the oviducts of pseudo-pregnant females, the pups were delivered, similar to controls [control sperm: 86 pups/325 embryos (26%), *Galntl5* KO sperm: 101 pups/202 embryos (50%)]. Thus, we conclude that the development of embryos fertilized with *Galntl5* KO sperm is normal. However, *Galntl5* KO sperm bound infrequently to ZP by insemination of cumulus-free eggs [binding sperm/egg: 14.3 ± 4.3 (Het), 1.3 ± 1.0 (KO)] (Fig. 3b, c). The decrease in sperm binding to ZP does not disrupt fertilization, but previous papers indicate a correlation between sperm ZP-binding and sperm migration through the UTJ[26,29,32,33,34,36,44,45]. To evaluate the ability of *Galntl5* null sperm to migrate through the UTJ, we observed the behavior of fluorescence-labeled *Galntl5* KO sperm in the female reproductive tract four hours after mating. While sperm from the control are observed in the oviduct, *Galntl5* KO sperm are abundant in the uterus and are rarely observed in the oviduct (Fig. 3d). Furthermore, when we focused on the UTJ region, *Galntl5* KO sperm bound to the UTJ are reduced compared to control sperm (Fig. 3e). Thus, *Galntl5* KO

sperm have a defect in UTJ binding and passage, leading to severe male fertility defects in vivo.

## GALNTL5 is a critical factor for UTJ migration

To reveal why *Galntl5* KO sperm show impaired UTJ migration, we examined the UTJ migration-required proteins in the TGC and sperm of *Galntl5* KO mice. So far, more than twenty genes expressed in the testis and epididymis have been identified as encoding essential factors for sperm migration into the UTJ, and ADAM3, a sperm membrane protein, commonly disappears in sperm lacking these UTJ migration-related proteins (referred to as the ADAM3-dependent pathway)[33]. Thus, we first examined ADAM3 in TGC and sperm of *Galntl5* KO males. Testicular ADAM3 is detected as a doublet band, and then ADAM3 is reduced in size by protein processing during epididymal transit[46]. ADAM3 in *Galntl5* KO TGC was detected at comparable levels with *Galntl5* Het TGC (Fig. 4a); however, the mature form of ADAM3 in *Galntl5* KO sperm was slightly decreased (Fig. 4b and Supplementary Fig. 4a). If the mature form of ADAM3 slightly exists in sperm, the impaired UTJ migration of *Adam3* KO sperm can be rescued in *Adam3* KO males with an *Adam3* transgene[45]. Therefore, we concluded that factors other than ADAM3 cause the impaired UTJ migration of *Galntl5* KO sperm. A previous paper showed that disruption of tACE led to an aberrant

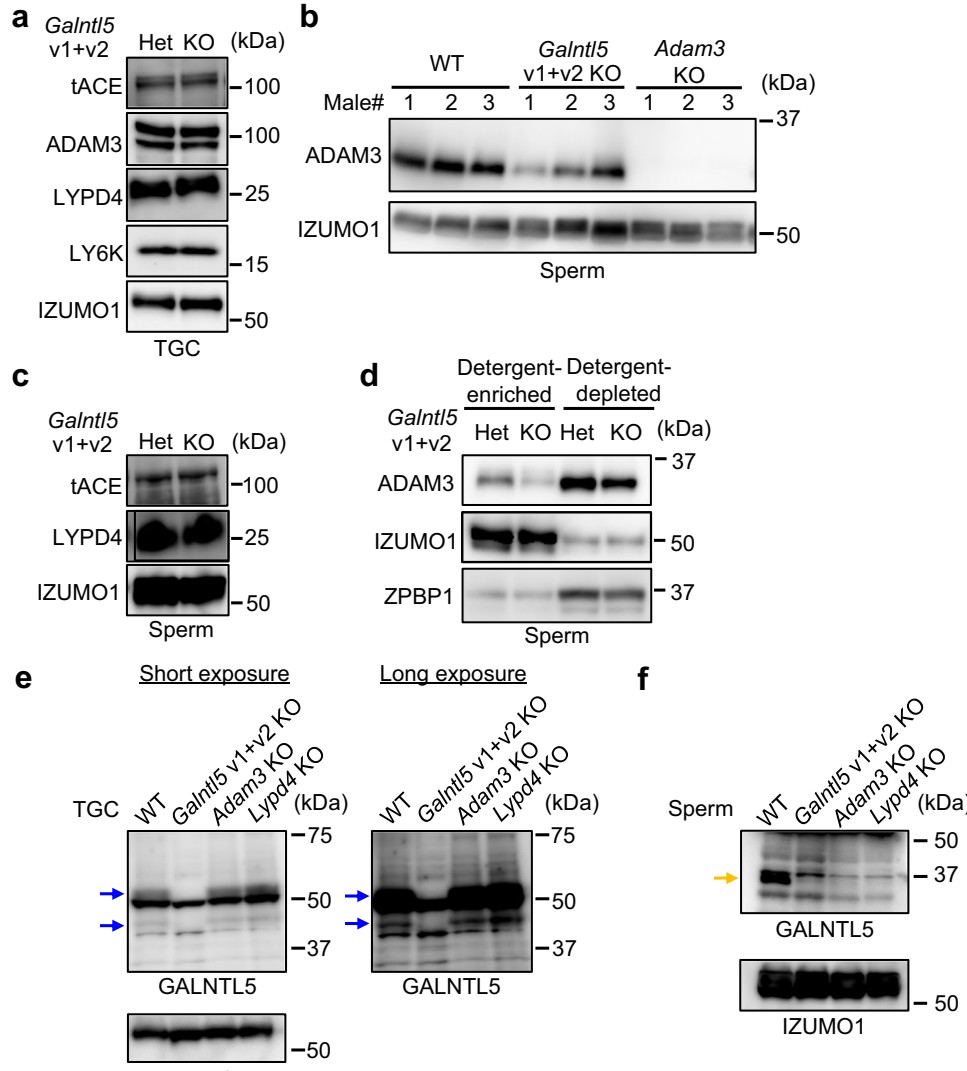

**Fig. 4 | GALNTL5 almost disappears in KO sperm of UTJ migration-related genes. a** Detection of testicular proteins required for sperm migration through the UTJ. IZUMO1 was used as the loading control. The data reproducibility was checked by two biological replicates. **b** Detection of sperm ADAM3. ADAM3 is slightly decreased in *Galntl5* KO sperm, but ADAM3 is absent in *Adam3* KO (also see Supplementary Fig. 4a). The data reproducibility was checked by five biological replicates. **c** Detection of sperm proteins required for sperm migration through the UTJ. The data reproducibility was checked by two biological replicates. **d** Observation of ADAM3 localization in sperm. IZUMO1 and ZPBP1 were used as the loading controls of the detergent-enriched and detergent-depleted phases. The data reproducibility was checked by two biological replicates. **e** Detection of testicular GALNTL5. Variants 1 and 2 (blue arrows) are detected in *Adam3* KO and *Lypd4* KO TGC. IZUMO1 was used as the loading control. The data reproducibility was checked by five biological replicates. **f** Detection of sperm GALNTL5. GALNTL5 almost disappears in *Adam3* KO and *Lypd4* KO sperm (yellow arrow) (also see Supplementary Fig. 4b). The data reproducibility was checked by five biological replicates.

distribution of ADAM3 in sperm, even if the mature form of ADAM3 was present in *tAce* KO sperm[27]. Thus, we examined tACE in *Galntl5* KO TGC and sperm, but we failed to detect a difference between *Galntl5* Het and KO males (Fig. 4a, c). Furthermore, to reveal the mature form of ADAM3 distribution in sperm, we separated sperm extracts from *Galntl5* Het and KO males into two fractions [detergent-enriched region (membrane proteins) and detergent-depleted region (soluble proteins)] using Triton X-114. The previous paper showed that the mature form of ADAM3 mostly disappeared in the detergent-enriched region from *tAce* KO sperm[27], indicating loss of ADAM3 from the sperm membrane. However, the mature form of ADAM3 could be detected in the detergent-enriched region from *Galntl5* KO sperm (Fig. 4d). These results indicate that ADAM3 is normally distributed in the *Galntl5* KO sperm membrane. From these analyses, we conclude that ADAM3-dependent mechanisms for sperm migration into the UTJ in *Galntl5* KO males are normal.

Previous papers showed that *Lypd4*, *Ly6k*, and *Pgap1* KO males are almost infertile because of impaired sperm migration through the UTJ, although these KO sperm have the mature form of ADAM3[34–36], suggesting there is an ADAM3-independent pathway to pass through the UTJ. Among the three factors, *Pgap1* is ubiquitously expressed in the multiple tissues, including the testis, based on the NCBI database (https://www.ncbi.nlm.nih.gov/gene/241062), but it remains unclear whether PGAP1 exists in sperm. Further, previous papers showed that PGAP1 functions as a GPI inositol-deacylase in the endoplasmic reticulum[37,38], suggesting that PGAP1 is not directly involved in the sperm migration into the UTJ and sperm-ZP binding. Thus, in this study, we examined LYPD4 and LY6K in *Galntl5* KO TGC and sperm. Previous papers showed that LY6K exists in only the testis[35] and LYPD4 exists in the testis and sperm[36,47]. The abundance of LY6K and LYPD4 in *Galntl5* KO TGC and sperm was comparable to *Galntl5* Het males (Fig. 4a, c). Therefore, we could not find significant differences in the

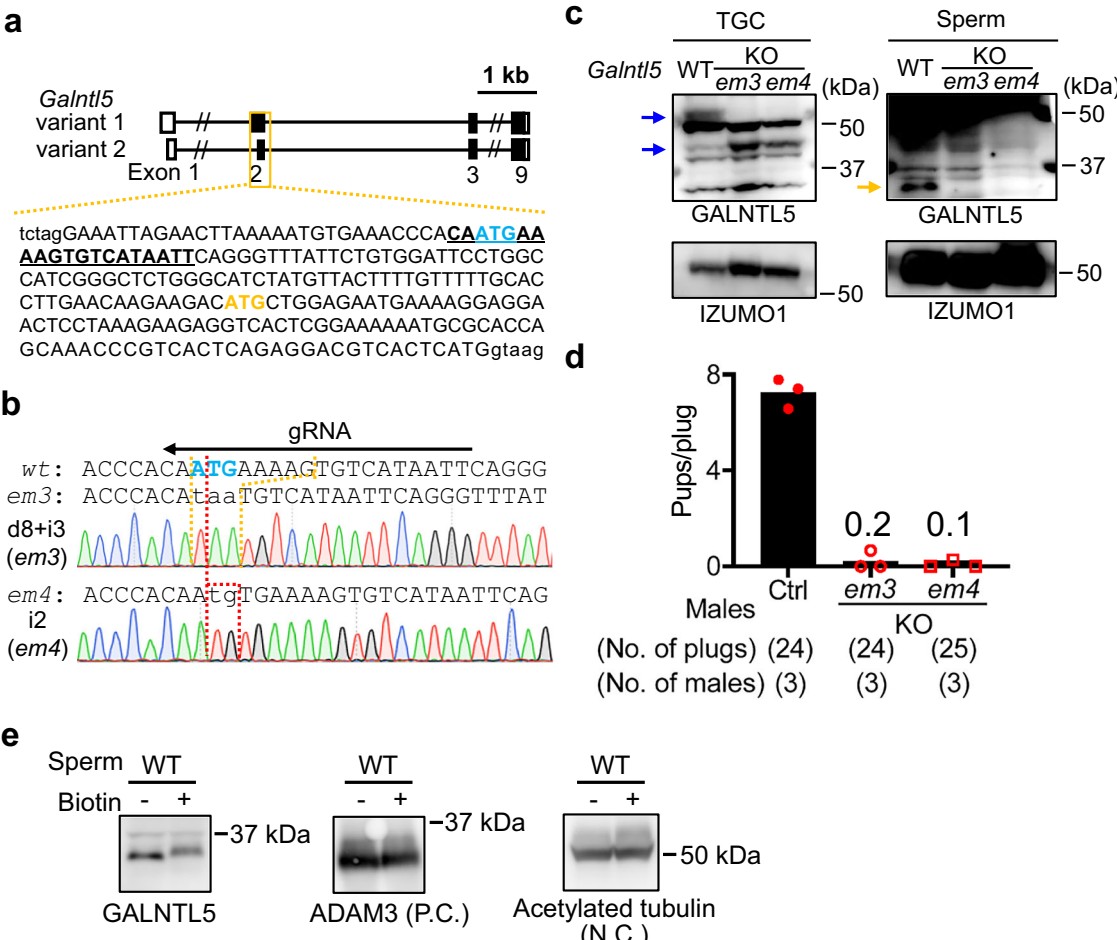

**Fig. 5 | GALNTL5 variant 1 is more important for male fertility. a** gRNA design to generate mice lacking *Galntl5* variant 1. Blue-colored and yellow-colored letters show the first methionine for variants 1 and 2, respectively. To delete the 1st Met for *Galntl5* variant 1, we designed a gRNA (underlined sequence). **b** Direct sequencing. The indel mutations were present in the *em3* (8 nt deletion and 3 nt insertion) and *em4* (2 nt insertion) alleles of *Galntl5*, leading to the disruption of the 1st Met for *Galntl5* variant 1 (blue-colored letters) (also see Supplementary Fig. 1c). The lowercase letters show the inserted nucleotides. **c** Detection of GALNTL5. Only the upper band, a protein from GALNTL5 variant 1, disappeared in the TGC of *Galntl5^em3* and *Galntl5^em4* KO mice. Furthermore, GALNTL5 at ~37 kDa disappeared in these KO sperm, confirming that only variant 1 of GALNTL5 was disrupted in *Galntl5^em3* and *Galntl5^em4* KO mice, and the processed protein from variant 1 existed in sperm.

Arrows show GALNTL5. IZUMO1 was used as the loading control. The data reproducibility was checked by two biological replicates. **d** Male fertility. *Galntl5^em3* and *Galntl5^em4* KO males are near sterile [pups/plug: 7.2 ± 0.6 (Ctrl), 0.2 ± 0.4 (*em3/em3*), 0.1 ± 0.2 (*em4/em4*)], and comparable to the male infertility of *Galntl5^em1* and *Galntl5^em2* KO mice. **e** Localization of GALNTL5 protein in sperm. If targeted proteins on the sperm surface are biotinylated, the molecular weight shifts up slightly. GALNTL5 was slightly shifted up, indicating that GALNTL5 is localized on the sperm surface. ADAM3 (sperm surface protein), and acetylated tubulin (cytoskeletal protein) were used for positive control (P.C.) and negative control (N.C.), respectively. The data reproducibility to detect GALNTL5 was checked by three biological replicates.

abundance and distribution of known proteins for UTJ migration between *Galntl5* Het and KO sperm. To reveal GALNTL5 in KO mice lacking UTJ-migration-related genes, we used *Adam3* KO and *Lypd4* KO male mice that demonstrate ADAM3-dependent and ADAM3-independent sperm migration[33]. Testicular GALNTL5 was detected in these KO TGC at a comparable level as in WT TGC (Fig. 4e), but sperm GALNTL5 nearly disappears in these KO sperm (Fig. 4f and Supplementary Fig. 4b). Thus, GALNTL5 is required for both ADAM3-dependent and ADAM3-independent sperm migration.

### *Galntl5* variant 1 is more important for male fertility

To reveal which *Galntl5* variant is more important for male fertility, we newly generated *Galntl5* mutant mice by introducing the gRNA/Cas9 protein complex into eggs to delete *Galntl5* variant 1 (Fig. 5a). By mating the F0 mutants with WT mice, we established two mutant lines; *em3* and *em4* have the indel mutation of 8 nt deletion and 3 nt insertion and the indel mutation of 2 nt insertion in exon 2 of the *Galntl5* variants, respectively, leading to the disruption of the first Met for *Galntl5*

variant 1 or the frameshift mutation (Fig. 5b and Supplementary Fig. 1a–c). As we obtained both KO mutant lines (*Galntl5^em3* and *Galntl5^em4* KO) by the intercross of Het mutants, we performed western blot analysis using extracts of the TGC and sperm and an antibody to recognize the C-terminus of GALNTL5. Only the upper band corresponding with predicted molecular sizes of variant 1 (~50 kDa) disappeared in the TGC of both KO mice, while the band at ~37 kDa also disappeared in the sperm of both KO mice (Fig. 5c). This result indicates that only variant 1 of GALNTL5 is disrupted in *Galntl5^em3* and *Galntl5^em4* KO mice, and the protein from variant 1 becomes the mature form during epididymal maturation. To reveal the effect of the disruption of *Galntl5* variant 1 on male fertility, control and *Galntl5^em3* and *Galntl5^em4* KO male mice were mated with WT females. The females mated with *Galntl5^em3* and *Galntl5^em4* KO males delivered pups infrequently (Fig. 5d and Supplementary Table 1), phenocopying *Galntl5^em1* and *Galntl5^em2* KO males (Fig. 1g). Thus, *Galntl5* variant 1 is more important for male fertility. Based on the TMHMM analyses, *Galntl5* variant 1 is predicted to encode a type II transmembrane protein, while

a protein encoded by *Galntl5* variant 2 lacks the transmembrane domain (Supplementary Fig. 1e). Further, the predicted protein from *Galntl5* variant 1 becomes the mature form lacking the putative transmembrane domain during sperm epididymal transit (Fig. 1f, Supplementary Figs. 1d and 3). To reveal whether sperm GALNTL5 exists on the sperm surface, we treated sperm surface proteins with a biotin labeling kit, and we observed an increase in the molecular weight of sperm GALNTL5 after biotinylation (Fig. 5e). Thus, the protein translated from *Galntl5* variant 1 exists on the sperm surface even after truncation during epididymal maturation.

## GALNTL5 attaches to the UTJ and ZP via binding with GalNAc

To reveal how GALNTL5 regulates sperm migration into the UTJ and sperm-ZP binding, we examined the physiological function of GALNTL5. As described in the introduction, GALNTL5 contains a conserved domain found in GalNAc transferase (also known as pp-GalNAc-T), suggesting that GALNTL5 may play a role in the glycosylation process in male germ cells. Although the previous publication did not demonstrate glycosyltransferase enzymatic activity of GALNTL5 in vitro[41,43], we examined whether the lack of GALNTL5 influences sperm glycosylation by western blot analysis with lectins [PNA (binding to Galβ3GalNAc in O-Glycan), MAL-II (binding to Neu5Acα3Galβ3GalNAc in O-glycan), LSL-N (binding to poly-LacNAc in O-glycan and N-glycan), and Con A (binding to αMan and αGlc in N-glycan)]. We failed to find overt differences in the glycosylation patterns between *Galntl5* Het and KO sperm (Fig. 6a). This result indicates that GALNTL5 does not have a function as a glycosyltransferase, corresponding with the previous papers[41,43]. As shown in Figs. 3b and 3e, *Galntl5* KO sperm bind weakly to the UTJ and ZP which contain sugars, such as mannose, N-acetylglucosamine, galactose, and GalNAc[48,49]. Thus, we examined the possibility that GALNTL5 directly binds to sugars in the UTJ and ZP. Specifically, we generated expression vectors that encode the amino acid sequences of ~50 kDa and ~37 kDa from the C terminus of GALNTL5 (Supplementary Fig. 5a) corresponding with the testicular and the expected sperm molecular sizes, and we subsequently obtained the protein lysates from the culture of cells transfected with these plasmids. Then, we incubated the lysates in sugar-binding gold nanoparticles or a GalNAc-immobilized gel (Fig. 6b, c). In the analysis using the sugar-binding gold nanoparticles, the binding of proteins to sugars leads to the precipitation of these particles, resulting in a decrease in the 530 nm absorbance. When we incubated GALNTL5 of ~50 kDa [testicular GALNTL5 (immature form)] with gold nanoparticles bound to mannose, GalNAc, or glucose, we could observe the decreased absorbance only for GalNAc nanoparticles (Fig. 6b). We further examined the binding of GALNTL5 proteins of ~50 kDa and ~37 kDa [expected sperm GALNTL5 (expected mature form)] and GalNAc using carbohydrate gel-equipped GalNAc-immobilized beads (Fig. 6c). The proteins that did not bind to GalNAc in the column were removed by washing. Proteins bound to GalNAc-immobilized beads were released by adding competing GalNAc; in this experiment, we could detect both immature and expected mature GALNTL5 proteins in the elution buffer (Fig. 6c). These results indicate that sperm GALNTL5 after testicular GALNTL5 is processed during epididymal maturation can also directly bind GalNAc. Furthermore, we examined which region of GALNTL5 is required to bind to GalNAc by narrowing down the amino acid sequence. As shown in Supplementary Fig. 5a, we generated three expression vectors encoding amino acid sequences of ~30, ~20, and ~10 kDa from the C terminal of GALNTL5, and then these vectors were transfected into the culture cells. After obtaining the cell lysates, we incubated the proteins with GalNAc-immobilized gel. We found that GALNTL5 of ~30 kDa and ~20 kDa (containing the predicted glycosyltransferase 2-like domain) can be detected in the elution buffer (Supplementary Fig. 5b). Our results indicate that at least 20 kDa from the C-terminal of GALNTL5 protein is required for GalNAc binding. Furthermore, to examine the GalNAc binding ability of human

GALNTL5 (Supplementary Fig. 5c), we incubated human GALNTL5 protein with the GalNAc-immobilized gel. As shown in Supplementary Fig. 5d, we found human GALNTL5 in the elution buffer, indicating that human GALNTL5 also has GalNAc binding ability. Based on these data, we conclude that mouse and human GALNTL5 can directly bind GalNAc.

To address how the binding of GALNTL5 and GalNAc regulates both sperm migration into the UTJ and ZP binding, we examined the distribution of GalNAc in mouse UTJ and ZP by lectin histochemistry with Dolichos Biflorus Agglutinin (DBA) lectin recognizing the terminal α-GalNAc. The specific immunofluorescence was observed in the epithelium of UTJ and ZP surfaces (Fig. 7a, b). To examine whether the blockade of GalNAc residues on the UTJ and ZP affects sperm binding, we treated the UTJ and unfertilized eggs with DBA, Wisteria floribunda (WFA), or Galanthus nivalis (GNL) lectins, which recognize the terminal α-GalNAc, GalNAc, and α-mannose, respectively, and then incubated these eggs with sperm. The blockage of terminal α-GalNAc on the UTJ using DBA lectin decreased the sperm number bound to the UTJ (Fig. 7c). While the blockage of mannose on the ZP failed to inhibit sperm-ZP binding, we could observe a significant decrease in sperm number bound to the ZP by blockage of GalNAc using DBA and WFA (Fig. 7d and Supplementary Fig. 6). Thus, sperm can bind to the UTJ and ZP surface through the recognition of GalNAc residues.

## Discussion

In our study, we independently established four mouse *Galntl5* mutant lines, and all KO male mice are sterile or have severe fecundity defects, unlike the previous reports. The phenotypic discrepancy may be caused by the different strategies to generate *Galntl5* KO mice. For example, *Galntl5* mutant mice registered on the IMPC database were generated by homologous recombination with a targeting vector (Supplementary Fig. 7a), and then the human beta-actin promoter-driven neomycin resistance cassette and exon 4 of *Galntl5* were deleted by the Cre-loxP system. Based on the database[50], these *Galntl5* mutant mice have lacZ expression in some mouse tissues, but *Galntl5* expression levels vary among mice. It is known that unpredicted transcripts sometimes occur due to the usage of splice acceptors other than the lacZ expression cassette. Thus, it is possible that individual differences in lacZ expression in *Galntl5* mutant mice are caused by leakage of lacZ expression, and the appearance of functional *Galntl5* transcripts by the change of alternative splicing patterns leads to normal fertility in these *Galntl5* lacZ KO male mice.

Takasaki et al. replaced the genomic sequences coding for exon 2 and part of intron 2 of the *Galntl5* gene with a neomycin resistance cassette to generate their *Galntl5* mutant mice[42]. However, *Gm55964*, a non-coding gene, exists in the intron region after exon 2 of *Galntl5* (Supplementary Fig. 7b), and a part of the sequence encoding *Gm55964* is deleted in their *Galntl5* mutant mice. In contrast to this previous paper[42], our established *Galntl5* mutant mice have the indel mutation only in exon 2. The expression in the male reproductive tract and the physiological function of *Gm55964* remain unclear, but the deletion of *Gm55964* may cause more severe male infertility in *Galntl5* mutant mice generated by Takasaki and his colleagues. As another possibility, we established *Galntl5* mutant mice on a B6D2 background, while Takasaki et al. established *Galntl5* mutant mice with E14 (129 substrain) embryonic stem cells. Thus, differences in mouse genetic backgrounds may cause phenotypic discrepancies in the *Galntl5* mutant males from our groups. Due to the poor reactivity of our antibody, we failed to obtain the specific signal of GALNTL5 by immunofluorescence staining of sperm. Previous studies have shown that ADAM3 and LYPD4, sperm proteins required for UTJ migration, are mainly localized in the anterior part of the sperm head and the outer acrosomal membrane, respectively[36,47,51,52]. Thus, we speculate that GALNTL5 also exists in the sperm head, because GALNTL5 is a

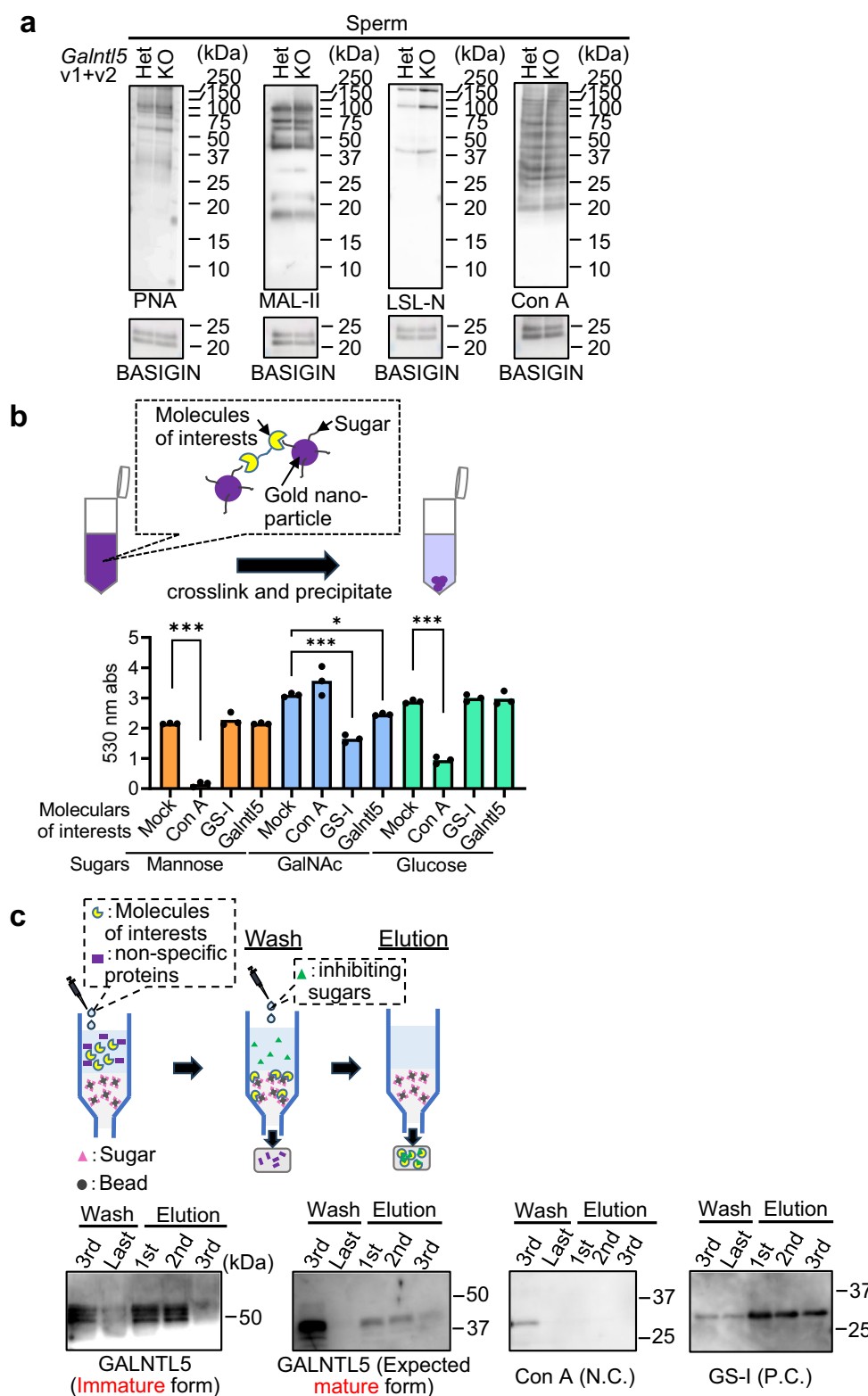

critical factor for sperm binding with UTJ and ZP. However, Takasaki et al. showed that GALNTL5 is concentrated in the neck region around the head-tail coupling apparatus of epididymal sperm[42]. Thus, connecting the phenotype of our KO mice with the reported GALNTL5 localization needs to be validated in future studies.

It is well-known that the maturation of sperm surface proteins by protein processing during epididymal maturation plays a critical role for

sperm to acquire fertilizing ability. For example, ADAM3, a sperm surface protein essential for sperm migration into the UTJ, becomes the mature form during sperm maturation in the epididymis, and impaired processing of ADAM3 causes a UTJ migration defect and male infertility[44–46,53]. Mouse *Galntl5* is transcribed from two alternative splicing variants (Fig. 1a), but human *GALNTL5* is transcribed from only one variant. Variant 1 with the transmembrane domain is conserved in both mice and humans

**Fig. 6 | GALNTL5 binds N-acetylgalactosamine (GalNAc). a** Lectin blot analysis. The pattern of glycan modifications in *Galntl5* KO sperm was comparable to Het sperm. BASIGIN was used as the loading control. The data reproducibility was checked by two technical replicates. **b** Binding assay of GALNTL5 and sugars using sugar-immobilized gold nanoparticles. By cross-linking the molecules of interest and sugar-immobilized gold nanoparticles, the complexes are precipitated, leading to a decrease in the absorbance at 530 nm. When the synthesized GALNTL5 protein was mixed with GalNAc-binding particles, a decrease in absorbance was observed (1-way ANOVA). *$p < 0.05$, ***$p < 0.01$. **c** Binding assay of GALNTL5 and sugars using carbohydrate gel. After adding the molecules of interest in the column equipped

with sugar-immobilized beads, only the molecules to specifically bind with the sugar are trapped in the beads. The trapped molecules are eluted by adding the inhibiting sugars. To examine the GalNAc-binding ability of GALNTL5, the proteins of ~50 kDa (immature form) and ~37 kDa (expected mature form) of mouse GALNTL5 (see Supplementary Fig. 5a) were incubated in a column equipped with GalNAc-binding beads. Both immature and mature GALNTL5 proteins were detected in the elution buffer. Con A and GS-I were used for N.C. and P.C., respectively. The data reproducibility to detect GALNTL5 was checked by two biological replicates.

(Supplementary Fig. 5c). Furthermore, we found that testicular GALNTL5 becomes the mature form during epididymal transit (Fig. 1f and Supplementary Fig. 3) and that sperm GALNTL5 is derived from variant 1 (Fig. 5c). Hence, we mainly focused on the physiological functions of variant 1 in this study and found that *Galntl5* variant 1 is necessary for male fertility (Fig. 5d). Although a putative function of *Galntl5* variant 2 on male fertility could be examined in future studies, our results indicate that the protein derived from *Galntl5* variant 1 mRNA on the surface of testicular germ cells is processed during epididymal maturation and is required for efficient binding of sperm to the UTJ and ZP and male fertility.

Our team has shown that disruption of several serine proteases, including ovochymase2 (OVCH2) specifically expressed in the caput epididymis, causes impaired ADAM3 processing and impaired sperm migration into the UTJ, leading to male infertility[46]. The previous paper showed that the S1 site in the catalytic pockets of human OVCH2 interacts with arginine (R) in the P1 site immediately before the cleavage site, which is a primary determinant of substrate specificity[54]. As shown in Supplementary Fig. 5c, mouse GALNTL5 has multiple arginine residues in the predicted cleavage site (as sperm GALNTL5 can be detected between 37 and 25 kDa, the cleavage site is thought to exist in this region). Thus, OVCH2 is a potential protease that cleaves GALNTL5 in the caput epididymis, but future studies to examine the interaction between OVCH2 and GALNTL5 are required to elucidate the GALNTL5 protein processing mechanism. While testicular GALNTL5 derived from variant 1 has the transmembrane domain, sperm GALNTL5 continues to exist on the sperm surface (Fig. 5e) via an unknown mechanism despite the lack of the transmembrane domain (Fig. 1f and Supplementary Fig. 1d). One putative mechanism is that other UTJ migration-related proteins on the sperm surface, such as ADAM3 and LYPD4, maintain GALNTL5 on the sperm surface as a complex; evidence for this hypothesis is that sperm GALNTL5 mostly disappears in *Adam3* KO and *Lypd4* KO sperm (Fig. 4f and Supplementary Fig. 4b).

Hamster sperm treated with some oligosaccharides, including GalNAc, barely bind to the ZP, indicating that sugar chains existing on the ZP are required for sperm-ZP binding[55]. Later studies revealed that O-glycans linked to mouse ZP3 are more important for sperm-ZP binding than N-glycans[10,11], and that the O-glycans attached to ZP3 have the terminal sequence with sialic acid, lacNAc (Galβ1-4GlcNAc), lacdiNAc (GalNAcβ1-4GlcNAc), Galα1-3Gal, and NeuAcα2-3[GalNAcβ1-4]Galβ1-4 (Sd[a] antigen) and are also shared with human ZP3[13]. The significance of α-GalNAc residues in sperm-UTJ binding was also reported in llama[49]; α-GalNAc residues were specifically detected in the UTJ of llama by the lectin histochemistry using DBA lectin, which recognizes the terminal α-GalNAc, and UTJ treated with DBA lectin had a decrease in sperm bound. Thus, the significance of GalNAc residues on the UTJ and ZP is becoming evident. Here, we reveal that blockade of GalNAc residues on the ZP and UTJ decreases sperm binding in mice (Fig. 7c, d and Supplementary Fig. 6), and that GALNTL5, an essential factor for both sperm migration into the UTJ and sperm-ZP binding, binds to GalNAc residues (Fig. 6b, c). Furthermore, we showed that more than 20 kDa of the C-terminus of GALNTL5 protein is required to bind GalNAc residues, and most of this region contains a glycosyltransferase 2-like domain (Supplementary Fig. 5). This domain is frequently observed in glycosyl transferases that transfer the sugar from UDP-glucose, UDP-N-acetyl-galactosamine, or

GDP-mannose to various substrates. Hence, we conclude that GALNTL5 on the sperm surface regulates sperm migration into the UTJ and sperm-ZP binding via binding to GalNAc O-linked oligosaccharides on the UTJ and ZP. Sperm bound to the UTJ can only migrate by swimming and beating when the luminal space is extended due to the oviductal contraction and UTJ relaxation[56]. Thus, we speculate that sperm bound to the UTJ through the interaction of GALNTL5 and GalNAc may be released from the UTJ and migrate into the oviduct by sperm movement and peristaltic activity of the uterus and oviduct.

Sperm-ZP binding is not limited to the same species. For example, porcine sperm can bind to bovine ZP, and the opposite also happens, indicating that there is little interspecies difference in sperm-ZP binding[57]. As described previously, the terminal α-GalNAc residues are also required for sperm-UTJ binding in llama[49]. Based on the Ensemble database, GALNTL5 is widely conserved in placental mammals (Supplementary Fig. 8). Furthermore, we also found that human GALNTL5 can bind to GalNAc (Supplementary Fig. 5d). Thus, we speculate that sperm migration into the UTJ and sperm-ZP binding through the binding of sperm GALNTL5 to GalNAc O-linked oligosaccharides on UTJ and ZP is an evolutionarily conserved mechanism among placental mammals, including humans. There are two possibilities as the reason GALNTL5 is required for the dual binding on the UTJ and ZP. Although a protein expression profiling of the UTJ has not been available so far, the O-linked glycans on ZP3, a specific protein on ZP, are required for sperm binding as described above[6,10]. Thus, GALNTL5 may recognize different O-linked glycoprotein targets in the UTJ and O-linked ZP3. As another possibility, there are unknown glycoproteins common to the UTJ and ZP, and GALNTL5 may recognize a unique shared glycoprotein.

## Methods

### Samples

Mice used in Japan for this study were purchased from Japan SLC or CLEA Japan. Mice were acclimated to a 12-h light/12-h dark cycle with ad libitum access to food and water. All animal experiments were approved by the Animal Care and Use Committee of Kumamoto University (A2021-035, A2021-168, and A2023-021) and Research Institute for Microbial Diseases at The University of Osaka (Biken-AP-H30-01). Human tissues were collected as nonhuman subject research by the Human Tissue Acquisition & Pathology (HTAP) Core at Baylor College of Medicine under the institutional review board (IRB) approved Protocol H-14435. Mice in Houston for tissue acquisition were maintained in accordance with NIH guidelines, and all animal procedures were approved by the Institutional Animal Care and Use Committee (IACUC) at Baylor College of Medicine (protocol AN-716).

### RT-PCR analysis

Mouse cDNA was prepared from multiple adult tissues of C57BL6/129SvEv hybrid mice and testes of 5- to 42-day-old mice. Human cDNA was prepared from multiple adult human tissues obtained from the HTAP Core. Total RNAs from these tissues were extracted using TRIzol reagent (Invitrogen, USA), and then cDNAs were synthesized from the total RNAs with SuperScript III Reverse Transcriptase (Invitrogen) following the manufacturer's instructions. The cDNAs were used for PCR with primer sets (Supplementary Tables 2 and 3).

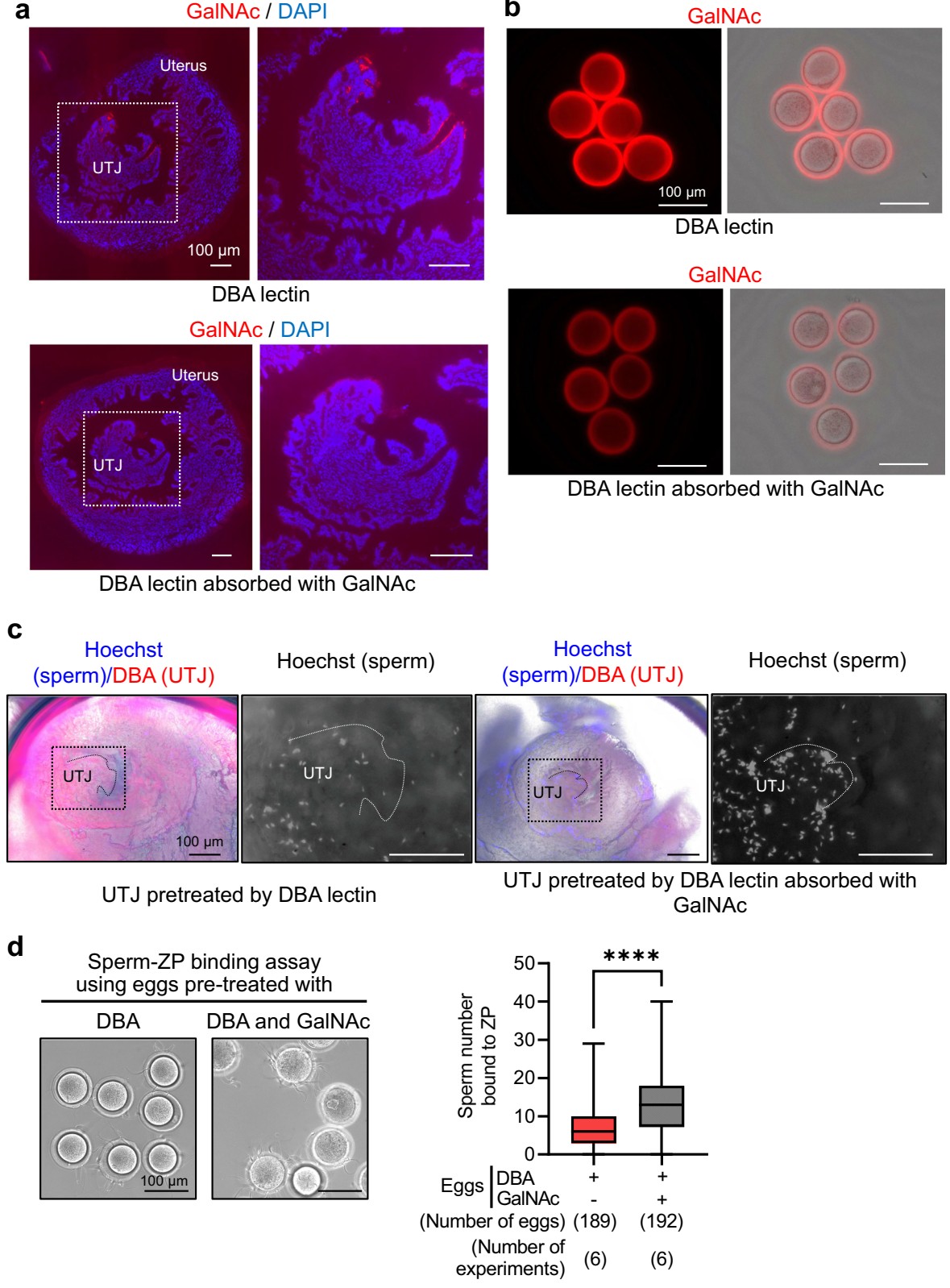

**a** GalNAc / DAPI

Uterus

UTJ

100 μm

DBA lectin

GalNAc / DAPI

Uterus

UTJ

DBA lectin absorbed with GalNAc

**b** GalNAc

100 μm

DBA lectin

GalNAc

DBA lectin absorbed with GalNAc

**c** Hoechst (sperm)/DBA (UTJ)    Hoechst (sperm)    Hoechst (sperm)/DBA (UTJ)    Hoechst (sperm)

UTJ    UTJ

100 μm

UTJ pretreated by DBA lectin

UTJ    UTJ

UTJ pretreated by DBA lectin absorbed with GalNAc

**d** Sperm-ZP binding assay using eggs pre-treated with

DBA    DBA and GalNAc

100 μm

****

Sperm number bound to ZP

50
40
30
20
10
0

Eggs | DBA : + / +
GalNAc : - / +
(Number of eggs) (189) (192)
(Number of experiments) (6) (6)

### Generation of *Galntl5* mutant mice

Pronuclear injection with gRNA/Cas9-expressing plasmid (for *em1* and *em2* mutations) and electroporation with the mixture of gRNA (Sigma-Aldrich, USA) and Cas9 enzyme (Nippon Gene, Japan) (for *em3* and *em4* mutations) were performed as previously reported[58–60]. Briefly, CARD hyperova (0.1–0.2 mL, Kyudo, Japan) was injected into the abdominal cavity of B6D2F1 females, followed by human chorionic gonadotropin (hCG) (5–7.5 units, ASKA Pharmaceutical, Japan). These females were mated with B6D2F1 males. After mating, the fertilized eggs with 2 pronuclei were collected from the oviductal ampulla. The gRNA sequences for *Galntl5* were "5'-AGGAGGAACTCCTAAAGAAG-3' (for *em1* and *em2* mutation)" and "5'-AATTATGACACTTTTCATTG-3' (for *em3* and *em4* mutation)". The injected or electroporated embryos were transported to the oviducts of pseudo-pregnant females. After 19 days, offspring

**Fig. 7 | Blockage of GalNAc on the UTJ and ZP surface decreased the sperm number bound to the UTJ and ZP. a** Detection of GalNAc in the UTJ. By the lectin histochemistry with Rhodamine-labeled Dolichos biflorus agglutinin (DBA) which recognizes the terminal α-GalNAc, DBA bound to the epithelium in UTJ. The Rhodamine-labeled DBA preabsorbed with GalNAc was used as the N.C. The area dotted by the white color was magnified (right panels). The data reproducibility was checked by three biological replicates. **b** Detection of GalNAc in the ZP. By live cell staining with Rhodamine-labeled DBA, DBA bound to the surface of ZP. The Rhodamine-labeled DBA preabsorbed with GalNAc was used as the N.C. The data reproducibility was checked by three biological replicates. **c** Sperm-UTJ binding assay. To evaluate the possibility of sperm-UTJ binding through GalNAc residues on the UTJ, the UTJ was incubated with DBA or DBA preabsorbed with GalNAc. After washing with TYH medium, the UTJ was inseminated with sperm pre-stained with Hoechst 33342. After fixation and washing, the sperm bound to the UTJ were observed. The Hoechst signal (the display changed to grayscale) in the area dotted by the black color was magnified (right panels). The data reproducibility was checked by four biological replicates. **d** Sperm-ZP binding assay. To evaluate the possibility of sperm-ZP binding through GalNAc on the ZP, eggs were incubated with DBA or DBA preabsorbed with GalNAc (see Supplementary Fig. 6). The eggs were washed with TYH medium and then inseminated with sperm for 30 min. After fixation and washing, the eggs were observed (left panel), and the sperm number bound to ZP was counted (right panel) (DBA: $6.8 \pm 5.3$, DBA + GalNAc: $13.4 \pm 7.5$) (Mann–Whitney test, $p < 0.0001$). Center line within the box: median, whiskers in box-and-whiskers plots: minimum to maximum, ****$p < 0.001$.

were obtained by Caesarean section or natural birth. The genotyping was identified by PCR with a primer and direct sequencing (Supplementary Tables 2 and 3). Then, KO mice were obtained by F1 × F1 intercrosses. After the F2 generation, mutants were used for phenotypic analyses. Frozen sperm from *Galntl5 em1* mutant males [B6D2-Galntl5<em1Osb>, RBRC# 10118, CARD# 2591], *Galntl5 em2* mutant males [B6D2-Galntl5<em2Osb>, RBRC# 11034, CARD# 2941], *Galntl5 em3* mutant males [B6D2-Galntl5<em3Kms>, CARD# 3517], and Galntl5 *em4* mutant males [B6D2-Galntl5<em4Kms>, CARD# 3518] will be available through RIKEN BRC (http://en.brc.riken.jp/index.shtml) and CARD R-BASE (http://cardb.cc.kumamoto-u.ac.jp/transgenic/).

### Antibodies
Information for all antibodies that were used in this paper is shown in Supplementary Table 4.

### Western blot analysis
Protein extracts from TGC and sperm were obtained as described previously[32]. Briefly, to collect the TGC, we disentangled the seminiferous tubules of the testis in PBS and then minced them using a razor. The suspension was filtered with a 59 μm nylon mesh filter. The sperm in the caput, corpus, and cauda epididymides were dispersed in PBS. After centrifugation, the TGC and sperm were homogenized in the cell lysis buffer containing protease inhibitor mixture (Nacalai Tesque, Japan) or boiled in the sample buffer. To obtain testicular sperm (TS), we disentangled the seminiferous tubules of the testis in PBS containing 1 mM EDTA, minced using a razor, and then filtered with a cell strainer with 70 μm mesh filter. After centrifugation at $600 \times g$, the pellets were suspended in 52% Percoll (Cytiva, USA), and then centrifuged with an ultracentrifuge (swinging bucket rotor SW 40 Ti, $26,000 \times g$, 10 min, 4 °C). The bottom layer was collected, incubated in ammonium-chloride-potassium (ACK) lysing buffer (Thermo Fisher) for the rupture of red blood cells, and then washed with PBS. After centrifugation, the TS was homogenized in the cell lysis buffer containing protease inhibitor mixture. To separate sperm membrane and soluble proteins, sperm proteins were extracted in TBS–Triton X − 114 (137 mM NaCl, 2.7 mM KCl, 25 mM Tris, 1% Triton X−114, pH 7.4) containing a protease inhibitor mixture at 4 °C for 30 min, and then incubated at 37 °C for 15 min. By centrifugation, we separated into the detergent-enriched phase (membrane proteins) and the detergent-depleted phase (soluble proteins). The protein extracts were separated by using sodium dodecyl sulfate-polyacrylamide gel electrophoresis (SDS/PAGE) (ATTO, Japan), and then transferred onto the PVDF membrane (Bio-Rad, USA). After blocking with 10% skim milk (Nacalai Tesque), the membrane was incubated with the antibodies. The HRP activities were visualized using ECL Prime (Bio-Rad) or Chemi-Lumi One Ultra (Nacalai Tesque).

### Natural mating
*Galntl5* mutant males (7–11 weeks old) were caged with two B6D2F1 females (7–9 weeks old) for more than one month. After separating from a male, we kept the females for another 20 days to check for potential delivery.

### Testis morphology and histological analysis
The testis morphology and weight from *Galntl5* mutant mice (15−39 weeks old) were examined. Histological analysis was performed as described previously[61]. Briefly, the testis was fixed in Bouin's fluid (Polysciences) at 4 °C overnight. The samples were dehydrated with increasing ethanol concentrations, and then were embedded with paraffin. The paraffin section was stained with 1% (w/v) periodic acid solution (Wako, Japan) for 10 min, followed by the treatment with Schiff's reagent (Wako) for 20 min, and then Mayer's hematoxylin solution (Wako) for 5 min. After dehydration with ethanol, these slides were mounted with Entellan® new (Merck, Germany) and observed under microscopy.

### Sperm morphology and motility
Sperm in the cauda epididymis were dispersed in PBS (for morphology) and TYH (for motility). Sperm morphology was observed using phase contrast microscopy. After 10 or 120 min of incubation, sperm motility was analyzed by CASA using our Ceros I analyzer as described previously[62].

### In vitro fertilization (IVF)
Pregnant mare serum gonadotropin (PMSG) (5−7.5 units, ASKA Pharmaceutical) was injected into the abdominal cavity of B6D2F1 females, followed by hCG (5−7.5 units) 48 h after PMSG injection. Then, eggs were collected from the oviductal ampulla of B6D2F1 females 12 h after hCG injection. For sperm binding assay to the zona pellucida, the cumulus cells of some eggs were removed by incubation with 33 μg/mL Hyaluronidase type IV-S (Sigma-Aldrich). Cauda epididymal sperm ($2 \times 10^5$ sperm/mL) pre-incubated in TYH medium for 2 h were inseminated with cumulus-intact and cumulus-free eggs. After 30 min of incubation, some eggs were fixed with 0.2% glutaraldehyde, and then the sperm number bound to the ZP was counted. The formation of pronuclei in the remaining eggs was observed 6 h after incubation.

### Sperm behavior in the female reproductive tract
For the visualization of sperm behavior in the female reproductive tract, hormone-treated B6D2F1 females were mated with *Galntl5* mutant mice with RBGS (red body green sperm) transgene that encodes EGFP in the sperm acrosome and DsRed in the mitochondria[63]. After 4 h of mating, the sperm behavior in the female reproductive tract was observed using confocal microscopy (Olympus, Japan) or fluorescence microscopy (Keyence, Japan). The observation of sperm bound to UTJ was conducted as described previously[64]. Briefly, the female reproductive tract collected from females after 1 h of coitus was fixed with 4% PFA, kept in acrylamide/bis-acrylamide solution [2.4 % (w/v) acrylamide, 0.8% (w/v) bis-acrylamide, and 0.25% (w/v) VA-044 initiator (Wako) in PBS] at 4 °C for 24 h, followed by the incubation at 37 °C for 3 h. The hydrogel-embedded female reproductive tract was shaken in 8% (w/v) SDS solution at 37 °C for 3 days, washed with PBS at 37 °C for 24 h, and then put in 88% (w/v) histodenz solution for 1 day. The fluorescent sperm in the female reproductive tract were observed under confocal microscopy.

## Biotinylation on sperm surface proteins

The biotinylation on the sperm surface proteins was conducted as described previously[65]. Briefly, sperm in the cauda epididymis were squeezed out, and incubated in PBS with or without 1 mM EZ-Link Sulfo-NHS-Biotin (Thermo Fisher, A39256) at room temperature for 30 min. After washing with PBS and centrifugation, TBS−Triton X−100 containing the protease inhibitor cocktail was added to the sperm pellet. The obtained sperm extracts were used for western blot analysis.

## Lectin blot analysis

Proteins from cauda epididymal sperm were extracted by TBS−Triton X−100 containing the protease inhibitor cocktail, and then used for the lectin blotting (Supplementary Table 4).

## Binding assay of GALNTL5 and sugars

The fused sequences of mouse *Galntl5* variant 1 ORF (ENSMUST00000030778) and affinity tag sequence (Flag or HA) were inserted under the CAG promoter (Supplementary Fig. 5a). Human *GALNTL5* variant 1 ORF (ENST00000392800) was amplified using Human Adult Normal Tissue First Strand cDNA-Testis (Wako) (Supplementary Tables 2 and 3), and then the amplicon and HA tag sequence were inserted under the CAG promoter. The analyzed all plasmids inserted human *GALNTL5* sequence have the mutation of p.Thr158Met (ACG → ATG), but this mutation was frequently observed based on gnomAD database (https://gnomad.broadinstitute.org/gene/ENSG00000106648?dataset=gnomad_r4). Thus, we used these vectors for the following experiments. The expression plasmids used for this assay will be available through Addgene [https://www.addgene.org/, pCAG1.1-*Galntl5* variant #1-3xHA (Addgene ID: 240645), pCAG1.1-*Galntl5* (37 kDa)-3xHA (Addgene ID: 240646), pCAG1.1-*Galntl5* (30 kDa)-3xHA (Addgene ID: 240647), pCAG1.1-*Galntl5* (20 kDa)-3xHA (Addgene ID: 240648), pCAG1.1-*Galntl5* (10 kDa)-3xHA (Addgene ID: 240649), and pCAG1.1-human *GALNTL5*-3xHA (Addgene ID: 241431)]. The plasmids were transfected into the culture cells (HEK293T and BMT10) by the calcium phosphate method or lipofectamine 2000 (Invitrogen). After 2 days of transfection, proteins from the culture cells were extracted with TBS−Triton X−100 containing the protease inhibitor cocktail. The binding assay with sugar-immobilized gold nanoparticles (SGNP; SUDx-Biotec, Japan) was done by following the manufacturer's instructions. Briefly, the culture cell proteins were incubated with FLAG M2 magnetic beads (Sigma-Aldrich) at 4 °C overnight to collect GALNTL5 protein. Then, GALNTL5 was incubated with SGNP, followed by the measurement of absorbance at 530 nm. GS-I (EY Laboratories, USA) and Con A (Mitsubishi Gas Chemical, Japan) were used as the control lectins. For the binding assay with a carbohydrate gel, the column containing GalNAc-immobilized acrylamide beads (EY laboratories) was washed with a buffer [0.01 M Tris-HCl, 0.15 M NaCl, 0.2 % (v/v) Triton X−100, pH 7.2], and then the column loaded with the cell lysates (200 μg) or lectins [200 μg, GS-I-biotinylated (EY Laboratories, BA-2401-2) and Con A-biotinylated (Vector Laboratories, B-1005)] was incubated at 4 °C for 30 min. After washing to remove the non-specific protein binding, the column was incubated in the buffer containing the inhibiting sugar [0.2 M N-Acetyl-D-galactosamine (Wako), 0.01 M Tris-HCl, 0.15 M NaCl, 0.2 % (v/v) Triton X−100, pH 7.2] at 4 °C for 30 min, and protein bound to GalNAc beads was eluted.

## Homology of amino acid sequences of mouse and human GALNTL5

The protein sequences of mouse and human GALNTL5 [UniPlot IDs: Q9D4M9 (mouse) and Q7Z4T8-1 (human)] were aligned with Clustal Omega (https://www.ebi.ac.uk/jdispatcher/msa/clustalo) (Supplementary Fig. 5c).

## Immunocytochemistry

The female reproductive tract was fixed with 4% PFA at 4 °C overnight, followed by the treatment with increasing sucrose to prevent ice crystals and the embedding with OCT compound (Sakura Finetek, Japan) as described previously[66]. To reveal the DBA distribution in the female reproductive tract, five-μm frozen sections were fixed with acetone and then incubated with blocking solution by following the manufacturer's protocol of the Glysite Scout Glycan Screening Kit (Vector Laboratories). These sections were incubated with Rhodamine-labeled DBA (final concentration: 20 μg/mL, Vector Laboratories) for 30 min, followed by sealing using VECTASHIELD Vibrance Antifade Mounting Medium with DAPI (Vector Laboratories). To reveal the DBA distribution in the unfertilized eggs, the cumulus cells of unfertilized eggs were removed using Hyaluronidase and then incubated with Rhodamine-labeled DBA (final concentration: 20 μg/mL) for 30 min. Rhodamine-labeled DBA preabsorbed with GalNAc (final concentration: 25 mM) was used as the negative control to check the specificity of DBA-positive cells.

## Blockage of the GalNAc residues on the UTJ and ZP

The UTJ and eggs were collected from hormone-treated females after 12 h of hCG injection as described previously. For the blockage of Gal-NAc residues on the UTJ, the UTJ was incubated in TYH medium containing Rhodamine-labeled DBA (final concentration: 20 μg/mL) or Rhodamine-labeled DBA preabsorbed GalNAc (final concentration: 25 mM) for 30 min, and then washed in TYH medium. For the sperm-UTJ binding assay, cauda epididymal sperm were dispersed in TYH medium for 10 min, stained with Hoechst 33342 for 10 min, and then diluted into $5 \times 10^6$ sperm/mL. After 10 min of dilution, sperm (final concentration: $5 \times 10^5$ sperm/mL) were inseminated with the UTJ for 30 min. For the blockage of GalNAc or Mannose residues on the ZP surface, the cumulus-free eggs were incubated in TYH medium containing Rhodamine-labeled DBA, Rhodamine-labeled DBA preabsorbed GalNAc, WFA-biotin (final concentration: 20 μg/mL), WFA-biotin preabsorbed GalNAc, or GNL-biotin (final concentration: 20 μg/mL) for 30 min. For the sperm-ZP binding assay, cauda epididymal sperm were incubated in TYH medium for 2 h, and then they (final concentration: $1 \times 10^5$ sperm/mL) were inseminated with cumulus-free eggs for 10 or 30 min. After fixation using 0.25% glutaraldehyde (Wako), the sperm binding to the UTJ and ZP was observed under a fluorescence microscope (Keyence) or a differential interference contrast microscope (Olympus).

## Statistics and reproducibility

All values are shown as the mean ± SD of at least three independent experiments. Significant differences were examined by the Mann−Whitney test (Figs. 2b, 3a, 3c, and 7d), the Kruskal−Wallis test (Fig. 2e and Supplementary Fig. 6), and 1-way ANOVA (Fig. 6b) using Prism9 (MDF, Japan).

## Reporting summary

Further information on research design is available in the Nature Portfolio Reporting Summary linked to this article.

# Data availability

Frozen sperm of the genetically modified mice established in this study are available from RIKEN BRC and CARD as described in the "Methods". Also, the expression vectors used in this study are available from Addgene. The data that support the findings of this study are available from the lead contacts as necessary. Source data are provided with this paper.

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

## Acknowledgements

This work was supported by the Ministry of Education, Culture, Sports, Science and Technology (MEXT)/Japan Society for the Promotion of Science (JSPS) KAKENHI grants [JP20H03172 and JP25K02196 to T.N.; JP25KJ1979 to A.T.; JP22H04922 (AdAMS) to K.A.; and JP21H05033, JP23K20043, and JP22H04922 (AdAMS) to M.I.]; Takeda Science Foundation grant to T.N. and M.I.; The Nakajima Foundation grant to T.N.; Senri Life Science Foundation grant to T.N.; The Inamori Research grant to T.N.; The Mochida Memorial Foundation for Medical and Pharmaceutical Research grant to T.N.; JST/PRESTO Grant JPMJPR2148 to T.N.; JST/CREST JPMJCR21N1 to M.I.; Japan Agency for Medical Research and Development (AMED) grant (JP23jf0126001) to M.I.; OU Master Plan Implementation Project promoted under Osaka University to M.I.; and the *Eunice Kennedy Shriver* National Institute of Child Health and Human Development (R01HD088412 to M.M.M. and M.I.). We thank Ms. Natsuki Furuta, Ms. Yoko Kimachi, Ms. Yumiko Moriwaki, Ms. Rina Nakamura, and Ms. Naoko Tanabe for their expert technical assistance, and Ms. Shirley Baker for help with manuscript formatting.

## Author contributions

T.N., R.U., Y.Z., K.A., M.M.M., and M.I. designed the research; T.N., R.U., D.M., H.S., Y.Q., A.T., R.M.M., D.T., M.N., and Z.Y. performed experiments; T.N., R.U., Y.Z., M.M.M., and M.I. analyzed the data; and T.N., R.U., M.M.M., and M.I. wrote the paper. All authors read and made comments on the submitted manuscript.

## Competing interests

The authors declare no competing interests.
