## [Peer Review file · Nature Communications]

GALNTL5 binds GalNAc and is required for migration through the uterotubal junction and sperm-zona pellucida binding

Corresponding Author: Dr Taichi Noda

Version 0:

Reviewer comments:

Reviewer #1

(Remarks to the Author)

This study is an impressive and well-executed exploration of the role of GALNTL5 in mouse sperm migration and fertilization. The authors' efforts in establishing and characterizing four independent Galntl5 null lines is commendable.

A major strength of this work is the combination of gene-editing, biochemical analysis, and binding assays, which together offer a comprehensive view of the role of GALNTL5 in sperm interaction with the while transiting through the UTJ and binding to the zona in vitro. The findings contribute significantly to one model of sperm migration through the UTJ and binding to the ZP, with potential relevance in comparative biology, given that GALNTL5 appears to be conserved among placental mammals. Overall, this study represents an important step forward in reproductive biology, offering compelling findings and setting the stage for future research in this area.

Comments:

The experiments presented in Figures 7C and 7D do not support the conclusion that "sperm can bind to the UTJ and ZP surface through recognition of terminal α -GalNAc residues." These results are based on competitive binding assays, which have previously been shown to oversimplify physiological interactions and may not fully capture their complexity.

The role of Galntl5 variant 2 should be experimentally investigated or, if beyond the scope of this study, explicitly discussed.

Reviewer #2

(Remarks to the Author)

This study investigates a gene expressed in the male reproductive tract that is essential for both sperm progression through the uterotubal junction (UTJ) and zona pellucida (ZP) binding on the oocyte. The identity of the sperm molecule that recognizes ZP components has been a longstanding gap in the field. This study demonstrates that the knockout of Galntl5, which encodes a sperm surface protein, impairs sperm binding to both the UTJ and ZP, resulting in severe male infertility. The researchers further show that GALNTL5 specifically binds to α -linked N64 acetylgalactosamine (α -GalNAc) present on the UTJ and ZP, proposing it as the key factor required for these crucial steps in fertility.

Major Comments

1. The difference between the two bands of GALNTL5 in testicular germ cell (TGC) lysates versus the 37 kDa band in sperm requires further explanation. Is there a specific protease or cleavage site responsible for the lower molecular weight in sperm? While this is discussed later, adding a sentence early on to introduce this possibility would improve clarity.
2. Figure 1: The data effectively support the text, but Figure 1F (right panel, C-terminal antibody) requires better explanation. A band near the specific band appears in the KO lanes but is not discussed. Additionally, in the control lanes, the bands detected in TGC lysates differ between the antibodies, which should not be the case. A more detailed description is needed to clarify these inconsistencies.
3. Figure 4: The western blot (WB) results for GALNTL5 show notable variation between blots. Can the authors explain these discrepancies? Are different sperm quantities loaded per lane?
4. If mature sperm are present in the TGC preparation, why is the 37 kDa band absent in these lysates?
5. The detection of GALNTL5 in testicular and epididymal sperm should be included. The authors have the necessary

antibodies, but if the signal is too non-specific, this should be acknowledged. Presenting this data would strengthen the manuscript.

6. Given that the 37 kDa sperm isoform lacks the transmembrane domain, could it be stripped from sperm and tested in binding assays?

7. What happens to embryo development when Galntl5 KO sperm fertilize wild-type eggs via IVF with cumulus-intact oocytes? Is plasma membrane binding disrupted in these sperm?

Minor Comments

• Lines 125–127: The following sentence needs clarification:

“So far, about 20 genes abundantly expressed in the testis and epididymis, such as a testicular isoform of angiotensin-converting enzyme (t-Ace), have been identified as essential factors for both sperm-ZP binding and UTJ migration. Most of these genes lack ADAM3 in sperm, but there are exceptions.”

Does this mean that knocking out these genes results in ADAM3 loss in sperm? Please revise for clarity.

• Lines 171–174: The phrasing should be improved for readability:

“To generate Galntl5 knockout (KO) mutants, we intercrossed heterozygous (Het) mutants. We then performed western blot analysis on testicular germ cell (TGC) and sperm extracts using antibodies recognizing both termini of GALNTL5 (Supplementary Fig. 1D) to confirm protein disruption in KO mice.”

• The functions of Lypd4, Ly6k, and Pgap1 should be briefly described in the Introduction or Discussion.

Overall Assessment

This manuscript presents compelling data with significant physiological implications. However, addressing the concerns above—especially those related to the biochemical properties of GAL-NTL5, WB inconsistencies, and sperm binding assays—would substantially strengthen the study.

Version 1:

Reviewer comments:

Reviewer #1

(Remarks to the Author)

The Authors have addressed my concerns.

Reviewer #2

(Remarks to the Author)

In this manuscript, the authors describe the identification of a molecule essential for both sperm progression through the uterotubal junction (UTJ) and the oocyte's binding to the zona pellucida (ZP). The identity of the sperm molecule that recognizes ZP components has been a longstanding gap in the field. This study demonstrates that the knockout of Galntl5, which encodes a sperm surface protein, impairs sperm binding to both the UTJ and ZP, resulting in severe male infertility. They also report α -linked N64 acetylgalactosamine (α -GalNAc) as the ligand of GALNTL5, and show that it is present on the UTJ and ZP, proposing it as the key factor required for these crucial steps in fertility.

The authors have responded adequately to all the raised points. In addition to a few minor typographical errors, there are some minor points worth highlighting.

1- The manuscript does not address the distribution of GALNTL5 in sperm. The authors indicate that this will be part of future studies, which is understandable. It is noted that OVCH2 might be the protease responsible for processing GALNTL5. Since a similar processing for ADAM3 is discussed, can the authors draw an analogy based on these previous studies regarding GALNTL5's location during female reproductive transit and possible changes between the UTJ and the ZP?

2- The authors indicate that fertilization is normal in cumulus intact oocytes but that in cumulus-free oocytes, binding is greatly reduced, ... "Galntl5-KO sperm bound infrequently to ZP by insemination of cumulus-free eggs." However, in the latter, fertilization is not disrupted under in vitro conditions. The present sentence does not make this clear, and the authors should state that lower binding does not disrupt fertilization under in vitro conditions.

3- The discussion is presented in a very cursory way, and could use more insights into the implications of the current findings. Some interesting points not addressed include why sperm have ~20 molecules whose deletion produces a similar phenotype of double binding to the UTJ and ZP? What are the mechanisms by which sperm that bind the UTJ through GALNTL5 are released from it?

Minor suggestions.

4- Line 222, better as ... Galntl5 KO sperm efficiently fertilized cumulus-intact eggs....

5-Line 298, when referring to double KO eggs, Galntl5em3 and em4, and should not be a superscript, here and throughout.

Reviewer #1 (Remarks to the Author):

Comments:

The experiments presented in Figures 7C and 7D do not support the conclusion that "sperm can bind to the UTJ and ZP surface through recognition of terminal α -GalNAc residues." These results are based on competitive binding assays, which have previously been shown to oversimplify physiological interactions and may not fully capture their complexity.

【Authors' response】

As pointed out by the reviewer, our experiments may not capture all phenomena in the sperm-UTJ binding and sperm-ZP binding, like the antibody inhibition experiment. However, since the molecular mechanism for sperm migration through the UTJ is unknown, we think it is meaningful to perform experiments examining the effect of blocking GalNAc residues using lectins on sperm-UTJ binding and sperm-ZP binding. Furthermore, in the revised experiment, we compared the effect of the blockage with DBA (terminal α -GalNAc-binding lectin), WFA (GalNAc-binding lectin), and GNL (α -mannose-binding lectin) on sperm-ZP binding. As you can see, DBA and WFA, but not GNL, can inhibit sperm binding to the ZP (**Supplementary Fig. 6**). Based on these data, we conclude that sperm could bind to the UTJ and ZP surface through recognition of the GalNAc residues. We added some sentences in the revised manuscript (lines 377-391, 690-709, and 1041-1065 of the manuscript with tracked changes), Supplementary Figure 6, and the information on the used lectins in the Supplementary Table 4. Also, we fixed

Figure 7A and C for the readers to clearly find the results and updated Figure 7D (because we performed the additional experiments).

Supplementary Figure 6. Sperm-ZP binding assay by blockage of GalNAc and mannose on the ZP surface. DBA, WFA, and GNL bind to terminal α -GalNAc, GalNAc, and α -mannose residues, respectively. DBA and WFA pre-absorbed GalNAc were used as the control (Ctrl). Eggs were pre-incubated in TYH drop with each lectin, washed, and then inseminated with sperm. After a 10-minute incubation, eggs were observed (panel A) and the sperm bound to the ZP were counted (panel B) (DBA: 6.1 ± 4.4 , DBA + GalNAc: 12.1 ± 5.2 , WFA: 5.8 ± 5.2 , WFA + GalNAc: 12.8 ± 6.6 , GNL: 14.6 ± 7.0). Center line within the box: median, whiskers in box-and-whiskers plots: minimum to maximum, ns: not significant, *: $p < 0.05$, ****: $p < 0.0001$.

The role of *Galnt15* variant 2 should be experimentally investigated or, if beyond the scope of this study, explicitly discussed.

【Authors' response】

Mouse *Galnt15* is transcribed from two alternative splicing variants, but human *GALNTL5* is transcribed from only one variant. Variant 1 with the transmembrane domain is conserved in both mice and humans. Furthermore, we found that sperm *GALNTL5* is

derived from variant 1 (**Figs. 1F and 5C**). Hence, we mainly focused on the physiological functions of variant 1 in this study. We wanted to examine whether variant 2 is required for male fertility in the future studies. This point was added to the discussion (lines 427-442 of the manuscript with tracked changes).

Reviewer #2 (Remarks to the Author):

Major Comments

1. The difference between the two bands of GALNTL5 in testicular germ cell (TGC) lysates versus the 37 kDa band in sperm requires further explanation. Is there a specific protease or cleavage site responsible for the lower molecular weight in sperm? While this is discussed later, adding a sentence early on to introduce this possibility would improve clarity.

Authors' response

To address the reviewer's comment, we collected testicular sperm and sperm from caput, corpus, and cauda epididymides and then examined the timing when GALNTL5 is processed from ~50 kDa to ~37 kDa using the antibodies to recognize the N- and C-termini of mouse GALNTL5. As shown in Figure 1F, the N-terminus antibody recognizes only the immature forms (~50 kDa and ~46 kDa), while the C-terminus antibody recognizes both immature and mature forms (~37 kDa). As shown in Supplementary Figure 3, when we used the N-terminus antibody, we could not detect the immature forms in epididymal sperm but only in testicular germ cells (TGC) and testicular sperm (TS). Using the C-terminus antibody, we barely found a specific signal in the TS due to the poor reactivity of the antibody, but we could detect the immature form in TGC and the mature form in sperm after the caput epididymis. Based on these data, we speculated that the immature form of GALNTL5 is cleaved after entering the caput epididymis by some proteases. Specifically, the disruption of a serine protease, such as ovochymase-2 (OVCH2) specifically expressed in the caput epididymis, causes the impaired ADAM3 processing and impaired sperm migration into the UTJ, leading to male infertility (Kiyozumi D et al., *Science*, 2020, PMID 32499443). The previous paper showed that the S1 site in the catalytic pockets of human OVCH2 interacts with arginine (R) in the P1 site immediately before the cleavage site which is a primary determinant of substrate specificity (Mehner C et al., *J Biol Chem*, 2022, PMID 35716777). As shown in Supplementary Figure 5C, mouse GALNTL5 has multiple arginine residues in the predicted cleavage site (as sperm GALNTL5 can be detected between 37 and 25 kDa, the cleavage site is thought to exist in this region.). OVCH2 may cleavage GALNTL5 in the caput epididymis, but future studies to examine the interaction between OVCH2 and

GALNTL5 are required to elucidate GALNTL5 processing mechanism. Thus, we added some text (lines 197-205, 444-454, and 550-559 of the manuscript with tracked changes) and new Supplementary Figures 3 and 5 in the revised manuscript.

Supplementary Figure 3. Detection of GALNTL5 in TGC, TS, and epididymal sperm. TGC (100 μ g), TS (40 μ g), and caput (Cap, 110 μ g), corpus (Cor, 110 μ g) and cauda (Cau, 50 μ g) epididymal sperm were used for SDS-PAGE. For the immunoblot analysis of GALNTL5, we used two antibodies to recognize the N- and C-termini of mouse GALNTL5 (see **Fig. 1F** and **Supplementary Fig. 1D**). We labeled immature (~50 kDa and ~46 kDa) and mature (~37 kDa) forms with blue and yellow arrows, respectively.

Using an N-terminus antibody, the immature forms were not detected in the epididymal sperm but in TGC and TS. Using a C-terminus antibody, we barely found any immature forms in TS due to the poor reactivity of this antibody. However, the immature forms were detected in TGC, and the mature form could be detected in epididymal sperm after the caput region. IZUMO1 was used as the loading control.

Supplementary Figure 5. Identification of the GalNAc binding region in GALNTL5.

C) Comparison of amino acid sequences of mouse and human GALNTL5. Blue colored letters show the matched amino acids between mice and humans. Based on the western blot data using testicular germ cells and sperm (see **Fig. 1F and Supplementary Fig. 3**), we speculate that GALNTL5 is cleaved between 37 kDa and 25 kDa from the C-terminal of GALNTL5 by some proteases. The serine protease “OVCH2”, which exists in the caput epididymis and is essential for sperm migration through the UTJ, recognizes arginine (R) for the cleavage of target proteins. Thus, arginine was shown by the red squares.

2. Figure 1: The data effectively support the text, but Figure 1F (right panel, C-terminal antibody) requires better explanation. A band near the specific band appears in the KO lanes but is not discussed. Additionally, in the control lanes, the bands detected in TGC lysates differ between the antibodies, which should not be the case. A more detailed description is needed to clarify these inconsistencies.

Authors' response

It is difficult to prevent the contamination of blood when TGCs are collected. Thus, the band at 50 kDa, heavy chain of IgG, was detected in both ctrl and KO TGC lanes using an antibody at the C terminus (Fig. 1F). The band patterns and intensity of the non-specific bands depend on the antibodies. As the readers easily find these bands as the non-specific bands, we have added red arrows at the right side of these bands. We showed the immature and mature forms of GALNTL5 as blue and yellow arrows, respectively. These sentences were added in the legend of Figure 1F.

Figure 1. Male mice disrupted for both variants of *Galntl5* are nearly sterile.

F) Detection of GALNTL5 proteins in testicular germ cells (TGC) and sperm. We used antibodies to recognize N- or C-termini of mouse GALNTL5 (**Supplementary Fig. 1D** and **Supplementary Table 4**). In the control (ctrl) TGC, two bands were detected at the predicted molecular sizes of variants 1 and 2 (~50 kDa and ~46 kDa, respectively) (see blue arrows) (also see **Supplementary Fig. 2** in which each antibody was incubated with the same PVDF membrane to detect GALNTL5). Using the C-terminus antibody, we also found non-specific bands of ~50 and ~37 kDa (red arrows) in both control and KO TGCs. In the ctrl sperm, GALNTL5 was mainly detected at ~37 kDa only when we used an antibody to recognize the C-terminus (see a yellow arrow). These specific bands disappeared in *Galntl5* KO TGC and sperm, indicating that GALNTL5 proteins from both variants were disrupted in these KO lines. IZUMO1 was used as the loading control.

As mentioned above, the band size of GALNTL5 variant 1 is similar to the IgG heavy chain. So, we performed a longer electrophoresis to detect GALNTL5 using an antibody against the C-terminus. Specifically, the protein amount (100 μ g) and gel concentration (5-20%) for the electrophoresis were the same condition, but we electrophoresed 100 and 105 minutes for the N- and C-terminus antibodies, respectively. This difference may lead

the discrepancy of non-specific band patterns detected in TGC lysates between the antibodies. To respond to the reviewer's comment, we detected GALNTL5 in the same PVDF membrane using both antibodies. Specifically, we firstly detected GALNTL5 using the antibody at N terminal. After stripping the N-terminus antibody from this membrane, we re-detected GALNTL5 using the antibody at the C-terminus. As shown in Supplementary Figure 2, the band size of GALNTL5 using the N-terminus antibody was comparable to the C-terminus antibody.

Thus, we added text in the legend of Figure 1F (lines 931-942 of the manuscript with tracked changes) and new Supplementary Figure S2 in the revised manuscript.

Supplementary Figure 2. Detection of GALNTL5 proteins in the TGC. Two control (#1-2) and three *Galntl5^{em2}* KO (KO #1-3) males were used to collect the TGC. TGC lysates (100 μ g) were used for SDS-PAGE. Testicular GALNTL5 was detected using the N-terminus antibody, and then the membrane, after stripping the N-terminus antibody, was re-probed with the C-terminus antibody. Blue and red arrows show immature forms and a non-specific band, respectively. IZUMO1 was used for loading control.

3. Figure 4: The western blot results for GALNTL5 show notable variation between blots. Can the authors explain these discrepancies? Are different sperm quantities loaded per lane?

【Authors' response】

As described in comment #2, the signal intensity of non-specific bands may vary even if the same protein amount is loaded due to blood contamination and the different timing of

sample collection. We had loaded the same protein amount (45 μ g for sperm) to all lanes, but there was still variation in the signal intensity of IZUMO1 in Figure 4B. Therefore, we performed western blot analysis using sperm (35 μ g) extracted on the same day (**Supplementary Fig. 4A**). Although ADAM3 protein intensity in the *Galntl5* KO sperm is reduced compared to WT sperm, it still remains in contrast to IZUMO1 and tubulin (loading controls) that are detected at comparable signal intensity in the WT and *Galntl5* KO sperm. This result is consistent with Figure 4B. Furthermore, we prepared new sperm lysates from WT, *Galntl5* v1+v2 KO, *Adam3* KO, and *Lypd4* KO males, and then performed western blot analysis (**Supplementary Fig. 4B**). Using our GALNTL5 antibody, we obtained a specific signal of \sim 37 kDa (yellow arrow) in WT sperm but not the other lanes, although the signal intensity of IZUMO1 and acetylated tubulin were comparable among samples. These results are similar to Figure 4F. Thus, we added Supplementary Figure 4 in the revised manuscript.

Supplementary Figure 4. Detection of ADAM3 and GALNTL5 in sperm lacking UTJ migration-related genes.

A) Detection of sperm ADAM3 in *Galntl5* KO sperm. The signal intensity of ADAM3 is reduced in *Galntl5* KO sperm, but ADAM3 remains (also see **Fig. 4B**). IZUMO1

and acetylated tubulin were used for loading controls.

- B) Detection of GALNTL5 in *Adam3* KO and *Lypd4* KO sperm.** The mature form of GALNTL5 was detected at ~37 kDa (yellow arrow), but the GALNTL5 signal disappears in *Adam3* KO and *Lypd4* KO sperm (see also **Fig. 4F**).

4. If mature sperm are present in the TGC preparation, why is the 37 kDa band absent in these lysates?

【Authors' response】

To address this reviewer comment, we performed western blot analysis using testicular sperm and sperm from each region of the epididymis. Please see comment #1 in response to reviewer critique #2 (**Supplementary Figure 3**). We can detect immature forms in TGC and TS, but cannot find the mature form in these germ cells. The mature form is detected in epididymal sperm from the caput region. Based on these data, we conclude that immature GALNTL5 is processed after entering the caput epididymis. Thus, we added some text (lines 197-205 of the manuscript with tracked changes) and generated new Supplementary Figure 3.

5. The detection of GALNTL5 in testicular and epididymal sperm should be included. The authors have the necessary antibodies, but if the signal is too non-specific, this should be acknowledged. Presenting this data would strengthen the manuscript.

【Authors' response】

As pointed out by the reviewer, we performed western blot analysis using testicular sperm and sperm from each region of the epididymis. Please see the comment #1 of the reviewer #2 (**Supplementary Fig. 3**). We found immature GALNT5 in testicular sperm, while mature GALNTL5 can be detected in the caput epididymis. Thus, we added some text (lines 197-205 of the manuscript with tracked changes) and made new Supplementary Figure 3.

6. Given that the 37 kDa sperm isoform lacks the transmembrane domain, could it be stripped from sperm and tested in binding assays?

【Authors' response】

To address the reviewer's comment, we incubated sperm protein extracts into a GalNAc-immobilized gel, and then examined whether the trapped GalNAc-binding proteins exist by CBB staining. However, we could not detect any bands in elution buffer, suggesting that it is necessary to optimize our protocol to bind sperm proteins with GalNAc-immobilized gel.

Thus, we changed the strategy to examine whether sperm GALNTL5 binds to GalNAc. Specifically, because sperm GALNTL5 is ~37 kDa, we generated an expression vector that encodes the amino acid sequence of the 37 kDa from the C terminal of GALNTL5 (**Supplementary Fig. 5A**). This plasmid was transfected into cultured cells, and then we incubated the cell lysates in a GalNAc-immobilized gel. As you can see, GALNTL5 of 37 kDa was detected in the elution buffer (**Fig. 6C**), suggesting that sperm GALNTL5 also has GalNAc binding ability.

Figure 6. GALNTL5 binds N-acetylgalactosamine (GalNAc).

C) Binding assay of GALNTL5 and sugars using carbohydrate gel. After adding the molecules of interest in the column equipped with sugar-immobilized beads, only the molecules to specifically bind with the sugar are trapped in the beads. The trapped molecules are eluted by adding the inhibiting sugars. To examine the GalNAc-binding ability of GALNTL5, the proteins of ~50 kDa (immature form) and ~37 kDa (expected mature form) of mouse GALNTL5 (see **Supplementary Fig. 5A**) were incubated in a column equipped with GalNAc-binding beads. Both immature and mature GALNTL5 proteins were detected in the elution buffer. Con A and GS-I were used for N.C. and P.C., respectively.

Furthermore, we examined which region from C-terminal of GALNTL5 is required to bind with GalNAc by narrowing down the amino acid sequence. As shown in **Supplementary Figure 5A**, we generated three expression vectors encoding amino acid sequences of 30, 20, and 10 kDa from the C terminal of GALNTL5, and then these vectors

were transfected into the culture cells. After obtaining the cell lysates, we incubated the proteins of 30, 20, and 10 kDa with GalNAc-immobilized gel. We found that GALNTL5 of ~30 kDa and ~20 kDa (containing the predicted glycosyltransferase 2-like domain) can be detected in the elution buffer (**Supplementary Fig. 5B**). Our results indicate that at least 20 kDa from the C-terminal of GALNTL5 protein is required for GalNAc binding. To examine the GalNAc binding ability of human GALNTL5, we incubated cell extracts transfected with a vector containing the human *GALNTL5* sequence with the GalNAc-immobilized gel. As shown in Supplementary Figure 5D, we found human GALNTL5 in the elution buffer, indicating that human GALNTL5 also has the GalNAc binding ability.

Supplementary Figure 5. Identification of the GalNAc binding region in GALNTL5.

- A) Plasmid construction.** Five expression vectors with amino acids of about 10, 20, 30, 37, and 50 kDa from the C-terminal of GALNTL5 were generated.
- B) GalNAc-binding ability of mouse GALNTL5 proteins.** GALNTL5 proteins were incubated in the GalNAc-immobilized gels (see **Fig. 6C**), and the obtained washed and elution buffers were used for western blot analysis. GALNTL5 proteins with more than 20 kDa containing the predicted glycosyltransferase 2-like domain could be detected in the elution buffer.
- C) Comparison of amino acid sequences of mouse and human GALNTL5.** Blue colored letters show the matched amino acids between mice and humans. Based on the western blot data using testicular germ cells and sperm (see **Fig. 1F and Supplementary Fig. 3**), we speculate that GALNTL5 is cleaved between 37 kDa and 25 kDa from the C-terminal of GALNTL5 by some proteases. The serine protease “OVCH2”, which exists in the caput epididymis and is essential for sperm migration through the UTJ, recognizes arginine (R) for the cleavage of target proteins. Thus, arginine was shown by the red squares.
- D) GalNAc-binding ability of human GALNTL5.** After incubating human GALNTL5 proteins in the GalNAc-immobilized gel, human GALNTL5 could be detected in the elution buffer.

Thus, we added some text (lines 345-376, 479-483, 493-494, 638-648, 669-672, and 1029-1039 of the manuscript with tracked changes), corrected Figure 6C, and added new Supplementary Figure 5 and the information on primers in the Supplementary Tables 2 and 3.

7. What happens to embryo development when *Galntl5* KO sperm fertilize wild-type eggs via IVF with cumulus-intact oocytes? Is plasma membrane binding disrupted in these sperm?

【Authors' response】

Galntl5 KO sperm can fertilize cumulus-intact eggs (**Fig. 3A**), indicating that the sperm-oolemma binding and fusion of *Galntl5* KO sperm are comparable to the control sperm. This result corresponds with the results of other UTJ-related factors including ADAM3. To examine the embryogenesis of eggs fertilized with *Galntl5* KO sperm, fertilized eggs obtained by incubation of *Galntl5* KO sperm and mutant (or WT) eggs were transferred into the oviducts of pseudopregnant females. The percentages of the delivered pups were

comparable to the control [control sperm: 86 pups/325 embryos (26%), *Galntl5* KO sperm: 101 pups/202 embryos (50%)]. Thus, we conclude that development of embryos fertilized with *Galntl5* KO sperm is normal. Thus, we added some text (lines 229-233 of the manuscript with tracked changes) in the revised manuscript.

Minor Comments

Lines 125–127: The following sentence needs clarification:

“So far, about 20 genes abundantly expressed in the testis and epididymis, such as a testicular isoform of angiotensin-converting enzyme (t-Ace), have been identified as essential factors for both sperm-ZP binding and UTJ migration. Most of these genes lack ADAM3 in sperm, but there are exceptions.”

Does this mean that knocking out these genes results in ADAM3 loss in sperm? Please revise for clarity.

【Authors' response】

We fixed this as shown in the following sentence (lines 131-132 of the manuscript with tracked changes).

“Most of these genes lack ADAM3 in sperm, but there are exceptions. -> Sperm with a knockout of most of these genes have ADAM3 loss, but there are exceptions.”

Lines 171–174: The phrasing should be improved for readability:

“To generate *Galntl5* knockout (KO) mutants, we intercrossed heterozygous (Het) mutants. We then performed western blot analysis on testicular germ cell (TGC) and sperm extracts using antibodies recognizing both termini of GALNTL5 (Supplementary Fig. 1D) to confirm protein disruption in KO mice.”

【Authors' response】

We fixed this as shown in the following sentence (lines 184-189 of the manuscript with tracked changes).

“As we obtained *Galntl5* knockout (KO) mutants by the intercross of heterozygous (Het) mutants, we performed western blot analysis using extracts of testicular germ cells (TGC) and sperm and antibodies to recognize both terminus of GALNTL5 (**Supplementary Fig. 1D**), to check the disruption of GALNTL5 proteins in KO mice. -> By intercrossing of heterozygous (Het) mutants, we obtained *Galntl5* KO mice. To check the disruption of GALNTL5 proteins in KO mice, we performed western blot analysis using extracts of testicular germ cells (TGC) and sperm and antibodies to recognize both terminus of

GALNTL5 (Supplementary Fig. 1D).”

The functions of *Lypd4*, *Ly6k*, and *Pgap1* should be briefly described in the Introduction or Discussion.

【Authors' response】

We added these explanations as shown in the following sentence (lines 132-146, and 276-286 of the manuscript with tracked changes).

“For example, our teams reported that ADAM3 persists in sperm lacking post GPI attachment to proteins 1 (*Pgap1*), lymphocyte antigen 6 complex, locus K (*Ly6k*), or Ly6/Plaur domain containing 4 (*Lypd4*), despite the findings that these KO sperm show a UTJ migration defect and impaired ZP binding³⁴⁻³⁶. Specifically, PGAP1 has a function as a GPI inositol-deacylase that removes the palmitate from inositol in the endoplasmic reticulum, suggesting that it is unlikely that PGAP1 is directly involved in the sperm migration into the UTJ and sperm-ZP binding^{37,38}. LY6K, a GPI-anchored protein, interacts with testis-expressed gene 101 (TEX101), which is a substrate of t-ACE in testicular germ cells^{35,39}. The role of LYPD4 in testicular germ cells and sperm in UTJ migration and ZP binding remains unclear, but this protein is present in *Adam3* KO sperm³⁶. Thus, we conclude that other sperm factors, rather than ADAM3, which is not conserved in humans, are responsible for sperm-ZP binding and sperm migration through the UTJ.”

“Previous papers showed that *Lypd4*, *Ly6k*, and *Pgap1* KO males are almost infertile because of impaired sperm migration through UTJ, although these KO sperm have the mature form of ADAM3³⁴⁻³⁶, suggesting there is an ADAM3-independent pathway to pass through the UTJ. Among the three factors, *Pgap1* is ubiquitously expressed in the multiple tissues, including the testis, based on the NCBI database, (<https://www.ncbi.nlm.nih.gov/gene/241062>), but it remains unclear whether PGAP1 exists in sperm. And, the previous papers showed that PGAP1 has the function as a GPI inositol-deacylase in the endoplasmic reticulum, suggesting that PGAP1 is not directly involved in the sperm migration into the UTJ and sperm-ZP binding^{37,38}. Thus, in this study, we examined LYPD4 and LY6K in *Galntl5* KO TGC and sperm. Previous papers showed that LY6K exists in only the testis³⁵ and LYPD4 exists in the testis and sperm^{36,47}.”

Overall Assessment

This manuscript presents compelling data with significant physiological implications. However, addressing the concerns above—especially those related to the biochemical

properties of GAL-NTL5, WB inconsistencies, and sperm binding assays—would substantially strengthen the study.

【Authors' response】

We addressed the concerns pointed out by the reviewer.

【Other #1】

In the revised experiments, we also used WFA lectin which also binds to both α - and β -GalNAc, and the pre-incubation with this lectin and cumulus-free eggs led to the decreased number of the sperm binding with ZP. Thus, we changed from α -GalNAc to GalNAc through the revised manuscript (lines: lines 66-67, 70, 73, 476, 485, and 495 of the manuscript with tracked changes).

【Other #2】

We replaced some images in Figures 3D, 4A, and 4C-D, because these images were low resolution. Furthermore, as we increased the experimental numbers, we updated Figures 1G and Supplementary Table 1.

【Other #3】

We fixed the incorrect information in Supplementary Figure 1A.

【Other #4】

We fixed the errors of grammar and words throughout the manuscript.

Reviewer #2 (Remarks to the Author):

Comments:

In this manuscript, the authors describe the identification of a molecule essential for both sperm progression through the uterotubal junction (UTJ) and the oocyte's binding to the zona pellucida (ZP). The identity of the sperm molecule that recognizes ZP components has been a longstanding gap in the field. This study demonstrates that the knockout of Galntl5, which encodes a sperm surface protein, impairs sperm binding to both the UTJ and ZP, resulting in severe male infertility. They also report α -linked N64 acetylgalactosamine (α -GalNAc) as the ligand of GALNTL5, and show that it is present on the UTJ and ZP, proposing it as the key factor required for these crucial steps in fertility. The authors have responded adequately to all the raised points. In addition to a few minor typographical errors, there are some minor points worth highlighting.

1- The manuscript does not address the distribution of GALNTL5 in sperm. The authors indicate that this will be part of future studies, which is understandable. It is noted that OVCH2 might be the protease responsible for processing GALNTL5. Since a similar processing for ADAM3 is discussed, can the authors draw an analogy based on these previous studies regarding GALNTL5's location during female reproductive transit and possible changes between the UTJ and the ZP?

【Authors' response】

As suggested by the reviewer, we added a few sentences regarding the speculation of GALNTL5 localization in sperm in the manuscript (lines 421-429 of the manuscript with tracked changes). Please see below.

“Due to the poor reactivity of our antibody, we failed to obtain the specific signal of GALNTL5 by immunofluorescence staining of sperm. Previous studies have shown that ADAM3 and LYPD4, sperm proteins required for UTJ migration, are mainly localized in the anterior part of the sperm head and the outer acrosomal membrane, respectively^{36,47,51,52}. Thus, we speculate that GALNTL5 also exists in the sperm head, because GALNTL5 is a critical factor for sperm binding with UTJ and ZP. However, Takasaki et al. showed that GALNTL5 is concentrated in the neck region around the head-tail coupling apparatus of epididymal sperm⁴². Thus, connecting the phenotype of our KO mice with the reported GALNTL5 localization needs to be validated in future studies.”

2- The authors indicate that fertilization is normal in cumulus intact oocytes but that in cumulus-free oocytes, binding is greatly reduced, ...” Galntl5-KO sperm bound infrequently to ZP by insemination of cumulus-free eggs.” However, in the latter, fertilization is not disrupted under in vitro conditions. The present sentence does not make this clear, and the authors should state that lower binding does not disrupt fertilization under in vitro conditions.

【Authors' response】

As it is for the readers to easily follow it, we added a sentence in the manuscript (lines 226-237 of the manuscript with tracked changes). Please see below.

“To assess the sperm fertilizing ability, we incubated *Galntl5* KO sperm with cumulus-intact and cumulus-free eggs. *Galntl5* KO sperm efficiently fertilized cumulus-intact eggs [fertilization rates: $96.9 \pm 2.7\%$ (Het), 100% (KO)] (Fig. 3A), indicating that *Galntl5* KO sperm show normal fertilizing ability *in vitro*. Furthermore, when these fertilized eggs were transferred into the oviducts of pseudopregnant females, the pups were delivered, similar to controls [control sperm: 86 pups/325 embryos (26%), *Galntl5* KO sperm: 101 pups/202 embryos (50%)]. Thus, we conclude that the development of embryos fertilized with *Galntl5* KO sperm is normal. However, *Galntl5* KO sperm bound infrequently to ZP by insemination of cumulus-free eggs [binding sperm/egg: 14.3 ± 4.3 (Het), 1.3 ± 1.0 (KO)] (Fig. 3B and C). The decrease in sperm binding to ZP does not disrupt fertilization, but previous papers indicate a correlation between sperm ZP-binding and sperm migration through the UTJ^{26,29,32,33,36,44,45}.”

3- The discussion is presented in a very cursory way, and could use more insights into the implications of the current findings. Some interesting points not addressed include why sperm have ~20 molecules whose deletion produces a similar phenotype of double binding to the UTJ and ZP? What are the mechanisms by which sperm that bind the UTJ through GALNTL5 are released from it?

【Authors' response】

As suggested by the reviewer, we added few sentences regarding the speculation of GALNTL5 localization in sperm in the manuscript (lines 489-494 and 505-511 of the manuscript with tracked changes). Please see below.

There are two possibilities as the reason GALNTL5 is required for the dual binding on the UTJ and ZP. Although a protein expression profiling of the UTJ has not been available so far, the O-linked glycans on ZP3, a specific protein on ZP, are required for sperm binding as described above^{6,10}. Thus, GALNTL5 may recognize different O-linked glycoprotein targets in the UTJ and O-linked ZP3. As another possibility, there are unknown glycoproteins common to the UTJ and ZP, and GALNTL5 may recognize a unique shared glycoprotein.

Sperm bound to the UTJ can only migrate by swimming and beating when the luminal space is extended due to the oviductal contraction and UTJ relaxation⁵⁶. Thus, we speculate that sperm bound to the UTJ through the interaction of GALNTL5 and GalNAc may be released from the UTJ and migrate into the oviduct by sperm movement and peristaltic activity of the uterus and oviduct.

Minor suggestions.

4- Line 222, better as ... Galntl5 KO sperm efficiently fertilized cumulus-intact eggs...

【Authors' response】

We fixed it (line 227 of the manuscript with tracked changes).

5-Line 298, when referring to double KO eggs, Galntl5em3 and em4, and should not be a superscript, here and throughout.

【Authors' response】

We wrote those according to guidelines for nomenclature of mutant mice (<https://www.informatics.jax.org/mgihome/nomen/gene.shtml>, please see the section 3.5.2), but we fixed some parts for the readers to easily follow it (line 306-307, 312, and 315-317 of the manuscript with tracked changes).

【Other】

According to the editorial requests, we fixed the manuscript.